# Pathophysiological Heterogeneity of the BBSOA Neurodevelopmental Syndrome

**DOI:** 10.3390/cells11081260

**Published:** 2022-04-08

**Authors:** Michele Bertacchi, Chiara Tocco, Christian P. Schaaf, Michèle Studer

**Affiliations:** 1Institute of Biology Valrose (IBV), University Côte d’Azur, 06108 Nice, France; chiara.tocco@unice.fr; 2Institute of Human Genetics, Heidelberg University, 69120 Heidelberg, Germany; christian.schaaf@med.uni-heidelberg.de

**Keywords:** BBSOAS, *NR2F1*, haploinsufficiency, neurodevelopmental disorder, genotype-phenotype correlation, clinical symptoms, mouse models

## Abstract

The formation and maturation of the human brain is regulated by highly coordinated developmental events, such as neural cell proliferation, migration and differentiation. Any impairment of these interconnected multi-factorial processes can affect brain structure and function and lead to distinctive neurodevelopmental disorders. Here, we review the pathophysiology of the Bosch–Boonstra–Schaaf Optic Atrophy Syndrome (BBSOAS; OMIM 615722; ORPHA 401777), a recently described monogenic neurodevelopmental syndrome caused by the haploinsufficiency of *NR2F1* gene, a key transcriptional regulator of brain development. Although intellectual disability, developmental delay and visual impairment are arguably the most common symptoms affecting BBSOAS patients, multiple additional features are often reported, including epilepsy, autistic traits and hypotonia. The presence of specific symptoms and their variable level of severity might depend on still poorly characterized genotype–phenotype correlations. We begin with an overview of the several mutations of *NR2F1* identified to date, then further focuses on the main pathological features of BBSOAS patients, providing evidence—whenever possible—for the existing genotype–phenotype correlations. On the clinical side, we lay out an up-to-date list of clinical examinations and therapeutic interventions recommended for children with BBSOAS. On the experimental side, we describe state-of-the-art in vivo and in vitro studies aiming at deciphering the role of mouse *Nr2f1,* in physiological conditions and in pathological contexts, underlying the BBSOAS features. Furthermore, by modeling distinct *NR2F1* genetic alterations in terms of dimer formation and nuclear receptor binding efficiencies, we attempt to estimate the total amounts of functional NR2F1 acting in developing brain cells in normal and pathological conditions. Finally, using the *NR2F1* gene and BBSOAS as a paradigm of monogenic rare neurodevelopmental disorder, we aim to set the path for future explorations of causative links between impaired brain development and the appearance of symptoms in human neurological syndromes.

## 1. NR2F1 as a Master Regulator of Brain Development and Function

Neurodevelopmental disorders (NDDs) of genetic origin are a highly heterogeneous group of pathological syndromic conditions caused by defects in the basic mechanisms of brain development, such as cell proliferation, migration and differentiation, as well as function and connectivity. The imbalance of these early developmental events can lead to structural and functional brain defects affecting cognitive functions, such as visuospatial processing, motor execution, learning and memory, attention and social skills. They can also represent an underlying risk factor for complex psychiatric conditions, including anxiety and depression.

A rare autosomal dominant neurodevelopmental disorder called Bosch–Boonstra–Schaaf Optic Atrophy Syndrome (BBSOAS), was first reported in 2014 on six patients [1] and then expanded over the past few years by clinicians all over the world (OMIM #615722). With an estimated prevalence between 1 in 100,000 to 250,000 people worldwide, BBSOAS has been so far diagnosed in more than 100 patients. However, new patients are reported every year, suggesting that this proportion could be an underestimation. BBSOAS symptoms are very heterogeneous and include optic nerve atrophy (OA) or optic nerve hypoplasia (ONH), cortical visual impairment (CVI), moderate to severe intellectual disability (ID), developmental delay, hypotonia, seizures, speech difficulties, motor dysfunctions, autism spectrum disorder (ASD) and others. It is the peculiar combination of these diverse symptoms, and particularly the CVI and OA, that differentiate BBSOAS patients from those affected by other NDDs with similar features. After the first description of BBSOAS, patients with *NR2F1* mutations from distinct cohorts could finally be grouped into a single clinical category under the name of a novel syndrome.

All BBSOAS patients identified to date present haploinsufficiency of the nuclear factor *NR2F1*, due mainly to de novo missense mutations or whole-gene deletion of only one of the two alleles. Nevertheless, some rare cases of inherited variants have been described. Complete *NR2F1* absence has never been reported to date, suggesting that this condition might be incompatible with life, as reported in mouse mutant models showing perinatal lethality upon loss of both *Nr2f1* copies [2]. *NR2F1* gene codes for a transcriptional regulator functioning in the form of a dimer and able to either activate or repress target gene expression, depending on the cellular and biological context [3,4,5]. Interestingly, pathogenic BBSOAS point mutations are principally located in the two most conserved functional domains of the protein: the DNA-binding domain (DBD), responsible for the interaction with target gene regulatory sequences, and the ligand-binding domain (LBD), necessary for dimerization and co-factor binding. Even when the same symptoms are shared by multiple BBSOAS patients, their severity can be variable, possibly depending on the specific type of *NR2F1* gene perturbation, suggesting the existence of a genotype-phenotype correlation.

The pivotal role of *NR2F1* for correct development is supported by its high degree of evolutionary conservation in the animal kingdom, with high protein homology between mice and humans. For this reason, mouse gain-of-function and loss-of-function models have been extensively used to explore the functions of this gene in brain development (reviewed in [5]). Mouse *Nr2f1* orchestrates different aspects of brain morphogenesis and maturation: acquisition of neocortical areal identity [6,7], control of proliferation and neurogenesis for laminar and cell-type specification in the cortex [6,8,9,10], proper hippocampal formation and synaptic plasticity [11,12,13], long-range migration of cortical interneuron subtypes [14,15], patterning of distinct domains in the developing eye and optic nerve [16,17], temporal control of neural progenitor competence [18,19] and activity-dependent shaping of pyramidal neuron morphology [20], among many other processes. The highly similar expression profile of mouse *Nr2f1* and human *NR2F1* suggests that the human gene could play similar roles [8,21,22,23,24]. However, whether the same functions have been preserved in human development is still unknown. Linking *Nr2f1* roles in mouse brain assembly with the clinical features described so far in BBSOAS patients could help in further converging diagnostic aspects on present and new patients, as well as unravelling the multi-faceted functions of this gene during brain development in both physiological and pathological conditions.

In this review, we have summarized the available clinical knowledge on individuals affected by BBSOAS, focusing on their predominant symptoms. We supply an updated list of both congenital and acquired BBSOAS features and propose a unified approach for early therapeutic intervention. By linking the clinical knowledge to the available experimental approaches, we revise the state of the art of cellular and animal models used to explore, both in vitro and in vivo, *Nr2f1* contribution to early brain development. Furthermore, focusing on the genotype–phenotype correlation that is starting to appear as new patients are reported, we describe the molecular functioning of NR2F1 dimerization and speculate on the possible pathogenic mechanisms originating upon *NR2F1* mutation, which could explain a dominant-negative effect during dimer formation. By highlighting what have been so far described in terms of cellular and molecular mechanisms and what is still missing to achieve a full comprehension of BBSOAS pathogenesis, we propose future experimental paths that could translate discoveries from the bench to the clinic.

## 2. From Distinct Cohorts to a Unified List of *NR2F1*-Haploinsufficient BBSOAS Patients

Due to the heterogeneity of symptoms, both in terms of presence and severity, a clear diagnosis of BBSOAS has been elusive until recently. In fact, long before *NR2F1* haploinsufficient patients were grouped in this syndrome, BBSOAS patients were included in autism, epilepsy or other clinical cohorts, depending on their most prominent pathological features. As an example, a patient with an *NR2F1* mutation in the DBD was initially identified in an epilepsy study on infantile spasms [25]. Similarly, a patient with prominent autism-like behaviors was reported after an unbiased search for de novo gene variants associated with autism [26]. The causative link between *NR2F1* haploinsufficiency and complex syndromic conditions remained unclear in the first clinical reports, as some of these patients were carrying large deletions also involving additional genes [27,28,29]. The first BBSOAS study, with patients selected from a cohort of individuals with CVI, finally reported loss-of-function point variants, proving once and for all the status of *NR2F1* as a disease gene and its causative link with BBSOAS symptoms [1,30]. Located on chromosome 5, the *NR2F1* gene is expressed in several tissues and organs, including the eye and the brain (Figure 1). Despite being initially described as an extremely rare syndrome, BBSOAS has now been diagnosed in more than 100 patients (Figure 1; complete list in Table 1). Novel cases are regularly reported and *NR2F1* is also recognized as an ASD candidate gene according to the Simons Foundation Autism Research Initiative (SFARI) database [31]. So far, a total of 112 *NR2F1* variants has been reported [32], and 92 BBSOAS patients have been clinically described, comprising 43 males and 46 females (Table 1). Most of the identified genetic variants have been diagnosed as de novo (73.9%), but some familial cases are also present (7.6%). Patients harboring other genetic variants or large deletions involving other genes are nonetheless included in Table 1. While 15 patients (16.3%) have small to large deletions involving *NR2F1* alone or together with adjacent genes, most of them (83.7%) harbor *NR2F1* point variants or small in-frame deletions. Among the latter group, 32 variants (34.8% of all patients) fall in the DBD, 17 (18.5%) in the LBD and 9 (9.8%) in the starting codon, with a single duplication variant falling out of the main functional domains. Additionally, eleven truncation and seven frameshift/truncation variants have been reported (12% and 7.6% of patients, respectively).

Each of the listed BBSOAS patients displays a peculiar combination of visual deficits with ID, DD, ASD and other neurological features that make them stand out as unique when compared to the clinical picture of other distinct neurodevelopmental syndromes. We describe such clinical phenotypes in detail in the next section.

## 3. Towards a Consensus List of *NR2F1*-Related Symptoms and Standardized Therapeutic Interventions for BBSOAS Patients

Initially identified for their optic nerve anomalies—notably optic atrophy— BBSOAS patients differentiate themselves from visually deficient patients with a combination of additional neurological symptoms creating a much more complex clinical picture (Figure 2), the heterogeneity of which represents a general hallmark of NDDs. The increasing number of BBSOAS patients reported to date allows the depiction of a general profile of common clinical features resulting from *NR2F1* loss-of-function. While the condition is mainly characterized by vision impairment (OA and other optic nerve abnormalities), CVI, developmental delay (defined as a delay in milestone acquisition in distinct domains, such as walking and first words) and ID [1], recent reports have revealed additional clinical characteristics, including hypotonia, infantile spasms and other forms of epilepsy, ASD or autistic traits, feeding difficulties due to oromotor dysfunction and hearing defects, among others [30,33,34,35,36,37,38,39,40]. Together with OA and CVI, the vision/eye phenotype additionally comprises ONH, manifest latent nystagmus/fusional maldevelopment and alacrima [38]. Going into the details of developmental delay and ID, most common features are low intelligence quotient (IQ), speech delay, behavioral abnormalities and fine motor skill impairment [38]. There are few exceptions to the speech delay, as some patients have been reported showing normal or even superior verbal abilities [1,33], suggesting that some cases of BBSOAS speech delay could depend more on impaired motor coordination than on intellectual disability *per se*.

Further, the more recent and comprehensive reports have also delineated additional features, such as high pain tolerance, remarkable long-term memory, prominent love for music, sleep difficulties and touch sensitivity [38], and novel brain morphological malformations, such as neocortical dysgyria/polymicrogyria in areas involved in language and number processing [8]. Moreover, a recent detailed evaluation of the visual deficits in BBSOAS patients suggested a stable, non-progressive reduction in visual acuity [41]. Finally, new, rarer symptoms were reported in isolated patients, such as mitochondrial deficit, psychosis, transient ataxia and protein-losing enteropathy [42,43,44]

Since the most common BBSOAS symptoms, either appearing at birth or developing during the first years of life, have now been described (Table 2), a panel of recommended clinical tests has been compiled by neuropediatricians for the follow-up of newly diagnosed individuals (Table 3), together with therapeutic approaches to be considered for early intervention (Table 4). However, as our knowledge on the phenotypic spectrum continues to grow, thanks to studies on larger groups of patients with *NR2F1* pathogenic variants [30,38], these lists are constantly updated and refined. Hence, families of recently diagnosed BBSOAS children should refer to up-to-date information available online at https://nr2f1.org (USA NR2F1 Foundation; URL accessed date: 17 March 2022) and https://nr2f1france.wordpress.com (French NR2F1 Association; URL accessed date: 17 March 2022).

In clinical practice, physicians should consider a diagnostic evaluation for BBSOAS in every person with ONH or OA in combination with developmental delay and ID. This implies that some of the key features of BBSOAS will only become evident over the course of the first years of life. During infancy, the symptoms of BBSOAS are less specific. Here, the diagnosis may be suggested in children with hypotonia, feeding difficulties, epilepsy and signs of visual impairment, such as poor eye tracking. Given that there is no pathognomonic feature of BBSOAS and considering the great genetic heterogeneity of NDDs such as BBSOAS, it goes without saying that the vast majority of individuals will be diagnosed based on genome-wide technologies: chromosome microarray analysis for deletions and whole-exome sequencing or large next-generation sequencing panels for single nucleotide changes and small indels (see Discussion).

Each of the main BBSOAS pathological features will be discussed in the following sections, focusing on their variability in both presence and severity, as well as cellular and molecular insights coming from in vitro and in vivo experimental models, which have advanced our understanding of NR2F1 function during brain development. Further, in a conclusive chapter, we will examine what is known so far about the correlation between genotype and phenotype in patients carrying different pathogenic *NR2F1* variants and speculate on the molecular mechanisms potentially impacting NR2F1 dimer formation and being responsible of this phenomenon.

## 4. Optic Atrophy (OA), Optic Nerve Hypoplasia (ONH) and Non-Progressive Reduction in Visual Acuity: BBSOAS as a Non-Canonical Optic Neuropathy

Inherited optic neuropathies are an important cause of visual impairment in young children with an estimated prevalence of 1 in 10,000 [45]. Although genetically heterogeneous with both nuclear and mitochondrial genes being implicated, the pathological hallmark is a pronounced vulnerability of retinal ganglion cells (RGCs), ultimately leading to ON degeneration and irreversible vision loss [45,46,47]. This phenomenon can be monitored by high-resolution optical coherence tomography (OCT). BBSOAS can be defined as a genetic optic neuropathy in light of its several and variable clinical features affecting the visual system, such as OA or optic nerve pallor, ONH or small optic discs, CVI, nystagmus (uncontrolled eye movements) and alacrima (decreased tear reflex) [1,30,38]. These features can be present alone or as comorbidities, with a severity degree that presumably varies depending on the type of *NR2F1* genetic perturbation.

One of the main and first-reported visual impairments in BBSOAS children is OA, as the name of the syndrome suggests. It can be defined as ON damage anywhere from the retina to the lateral geniculate nucleus of the thalamus, usually caused by RGC death and retraction of RGC axons, with a resulting pale ON in fundoscopy. Notably, OA is particularly common in patients bearing DBD variants (78%), translation initiation variants (78%) and frameshift/truncations (72%), while it is less represented in milder cases caused by LBD variants (47% of patients). However, since the first description of BBSOAS patients, OA and ONH have both been reported [1]. As OA, ONH is also characterized by a deficiency of RGCs and their axons, leading to ganglion cell layer disorganization and a small optic disc with a thin ON. ONH is often associated with poor fixation, abnormal eye movements, nystagmus, strabismus, hyperopia [48] and vision ranging from no light perception to good functional vision [49,50]. Various theories have been proposed to explain the etiology of ONH, including a developmental failure of RGCs [51,52,53,54]. The description of ONH may sometimes be confused with OA, and their overlap may lead to diagnostic challenges [55,56,57].

A main difference between ONH and OA is that while ONH is a congenital, non-progressive disease characterized by underdevelopment of the ON, OA is instead degenerative with a normal early development of the ON that then deteriorates over time. As an example, in autosomal dominant or recessive OA caused by pathogenic *OPA1* (OMIM 605290) and *WFS1* (OMIM 606201) variants, progressive RGC loss starts in early childhood, and most patients are registered as legally blind by the fifth decade of life [58]. Furthermore, ONH is often syndromic, and occurs in conjunction with other neurodevelopmental abnormalities, such as brain malformations, developmental delay, ID and/or ASD [59,60,61]. The most common neuroanatomical malformation found in patients with ONH is hypoplasia of the corpus callosum, associated with developmental delay, neurological deficits and seizures [60,62], all clinical features also characteristic of BBSOAS children [3,38]. Genetically, syndromic ONH can be caused by perturbations of key transcription factors orchestrating brain, eye and ON development, such as SOX2, PAX2 and PAX6 [63,64,65]. In summary, while ONH has a developmental origin and is often syndromic, stable and non-progressive, OA is instead a progressive, i.e., degenerative, RGC axonal loss due to both genetic and environmental reasons.

Experimental studies in *Nr2f1*-deficient mouse models can help in filling the lack of clinical data due to the rarity of BBSOAS patients (Figure 3) and contribute to discerning OA from ONH in a BBSOAS-like context. Thanks to a high degree of evolutionary conservation [66], mouse *Nr2f1* and human *NR2F1* are similarly expressed in different retinal cell types, RGCs included [16], allowing the use of constitutive mouse models with *Nr2f1* haploinsufficiency as a model to study BBSOAS-related visual impairments [3,16]. Mouse *Nr2f1* retinal expression controls the density of RGCs, which is reduced in mutants due to a differentiation delay during embryogenesis and excessive apoptosis around the time of birth [16,41]. The early decrease in RGCs ultimately affects the number of axonal fibers in the ON, resulting in OA. Besides being atrophic, mutant ONs also show low levels of myelination due to a delay in oligodendrocyte migration and maturation at both embryonic and post-natal stages [16,67]. Interestingly, the defect appears to be restored upon treatment with Miconazole, a chemical drug promoting oligodendrocyte proliferation and maturation [16,68,69,70,71,72], thus proposing a possible therapeutic approach. Mouse mutant ONs are further impacted by an inflammatory process involving ON astrocytes at embryonic stage and in early post-natal life [16]. This early ON inflammation is of particular interest, as it could impact RGC survival and/or worsen myelination defects, as it happens in inflammatory forms of human optic neuritis [73,74]. In a reinforcing feedback loop, hypomyelination could in turn increase the loss of ON axonal fibers and retinal RGC apoptosis [75]. Likely due to these multiple concurrent and early developmental causes (inflammation, hypomyelination, RGC apoptosis and loss of RGC axons), a significant delay in the conduction velocity of visual stimuli along the ONs of mutant animals was revealed by electrophysiological recordings [16]. RGC cells apart, the development of other retinal cell types, such as cone photoreceptors [76], amacrine and bipolar cells [77], seems also to be *Nr2f1*-dependent, suggesting that its dysfunction could affect color vision and visual acuity by impacting on distinct retinal cell types.

In summary, according to experimental data in *Nr2f1*-deficient mouse models, BBSOAS-like visual impairments, such as retinal and ON anomalies, result from primary RGC patterning, myelination and inflammatory defects originating during early development (Figure 3) [16,41], and leading to visual system defects that would remain stable during postnatal life [41]. Consistently with early appearance of BBSOAS, ONH could be the major clinical feature accounting for the non-progressive visual dysfunction observed in patients [41]. We propose that since BBSOAS differs from classical optic neuropathies, such as OA, in which vision loss is progressive and visual acuity decreases in time, this syndrome should be defined as a non-progressive ONH with early (embryological) developmental origin. However, more studies in a higher number of patients and during a longer follow-up period will be necessary to confirm the stationary nature of visual clinical features, as current reports cannot exclude a progressive worsening later in a patient’s lifespan.

## 5. A Genetic Control of Optic Disc Abnormalities in BBSOAS

Several BBSOAS patients show small malformations of the optic disc (OD) at the edge between the neural retina and the ON. Such malformations manifest in excavated, pale or small ODs. Often associated with other congenital eye malformations, OD lesions can negatively affect visual acuity. Interestingly, OD malformations are reported more frequently in patients with *NR2F1* deletions (33%) than in patients with other variants (12–22%). A structural change in the form of pale and/or excavated ODs is diagnosed by ophthalmologists as an indirect sign of underlying ON diseases, including OA or ONH. In fact, OD pallor and abnormal shape can result from RGC death (causing axonal fibers loss) and degeneration of pial capillaries entering the optic nerve head [57,78,79]. However, some OD malformations—such as optic disc coloboma—can be the direct consequence of genetic or environmental insults affecting the development of this region, rather than a consequence of ON disease [80,81,82]. The OD pit, for example, consisting in a round or oval localized depression within the OD, is caused by defective occlusion of the embryonic ventral fissure of the optic vesicle [83,84]. By homology, data from *Nr2f1* mouse models showed delayed ventral fissure closure in *null* embryos [16] or severe coloboma in double *Nr2f1* and *Nr2f2* conditional mutants [17], suggesting that impaired ventral fissure fusion could be responsible for OD malformations in a *Nr2f1*-deficient context. This would also fit with reports of BBSOAS patients showing cases of coloboma-like malformations [29,30,85].

Furthermore, data in mice revealed a molecular network controlled by *Nr2f1* and driving the establishment of the border between the Pax2-expressing optic stalk (i.e., the presumptive ON) and the Pax6-expressing retinal vesicle (i.e., the presumptive neural retina), ultimately allowing the development of the OD between these two regions (Figure 3) [4,17]. One key function of the OD is to produce signaling molecules for RGC axonal guidance, such as Netrin-1 [86,87]. *Nr2f1* constitutive mutants demonstrated how the OD genetic network is disrupted by loss of *Nr2f1* alone, resulting in *Pax6* overexpression at the expense of the Pax2 domain [16]. This results in a shift of the border between NR and OS, with heavy consequences for *Netrin-1* expression, ultimately impacting axonal guidance of RGC fibers exiting the eyeball [41]. However, defects in *Nr2f1* haploinsufficient animals appear quite subtle compared to OD malformations in BBSOAS patients, suggesting that additional factors could impact OD development in humans and/or that species-specific differences are present.

The involvement of *Pax2* in an *Nr2f1*-regulated network is particularly important, as human *PAX2* mutations lead to coloboma-like OD malformations in the context of syndromic conditions [81,88,89], consistent with a key role of the *Nr2f1-Pax2-Pax6* genetic network for the correct establishment of the OD region. Human *NR2F1* could indeed control OD development by modulating *PAX2* and *PAX6*, as it does in mouse, but further experiments on human cells are necessary to verify the evolutionary conservation of such a network.

## 6. When Intellectual Disability Meets Visual Disfunction: Cortical Visual Impairment (CVI) in BBSOAS Patients

The condition of CVI (affecting around 42% of BBSOAS patients, and particularly common in the ones bearing DBD variants) is a bilateral visual impairment due to a non-ocular cause in the presence of normal pupil reactivity and characterized by abnormal perception, elaboration and interpretation of visual stimuli [90,91]. Despite the presence of nystagmus, ON atrophy and other structural eye anomalies, the degree of visual impairments in BBSOAS patients with CVI exceeds what would be expected from eye abnormalities alone, implying that higher-order visual centers in the brain (such as the retro-chiasmatic visual pathways, the thalamus, the primary cortex and/or the secondary associative visual cortices) might also be affected [38,91,92,93,94,95,96,97,98]. In BBSOAS patients, poor visual acuity and visual field abnormalities support the diagnosis of CVI. For instance, patients have narrowed visual fields, low visual acuity, and difficulties with distance viewing, following fast moving scenes or recognizing objects in crowded environments (Table 2). Among the possible factors contributing to CVI in BBSOAS, impaired thalamic connection with parietal and occipital cortices has been hypothesized to impact visuospatial ability [33], together with aberrant connections of major fasciculi relaying the occipital lobe to adjacent associative areas [41].

*Nr2f1* mouse mutant models challenged the involvement of central thalamic and neocortical structures at the anatomical, electrophysiological and behavioral levels (Figure 3). The size of visual thalamic nuclei is affected upon *Nr2f1* loss, in turn acting on the maturation of primary and secondary visual areas as a result of impaired afferent thalamocortical innervation [2,7,99]. At cortical level, the *Nr2f1* high caudolateral to low rostromedial expression gradient is key for a dose-dependent establishment of distinct neocortical areal identities, a process termed arealization [100,101]. Upon mouse *Nr2f1* loss, the rostral motor area expands at the expense of caudal sensory ones, so that the primary visual area (V1) tends to be compressed caudally in *Nr2f1* mutant neocortex, with a striking effect in *null* (homozygous) brains and a less pronounced area shift in heterozygous *HET* animals [2,7,66]. Electrophysiological recordings demonstrated a delay in transmitting peripheral visual stimuli along the visual pathway, together with a decrease in amplitude in the visual thalamic nucleus, the superior colliculi and the V1 superficial cortical layers of *HET* mice [16,41]. Suboptimal conduction velocity of visual stimuli along the visual pathway could depend on ONH associated with hypomyelination and gliosis [16]. Furthermore, visual acuity is decreased in *Nr2f1 HET* mice compared to *wild-type* littermates, in line with low visual acuity and nondegenerative vision loss in BBSOAS patients [1,30,38,41]. Finally, CVI could also result from impaired elaboration of visual information in secondary associative cortices. Interestingly, while *Nr2f1* haploinsufficient mice are still able to learn and execute complex tasks [12,16], they nevertheless fail to associate the visual stimulus with a specific operating task when challenged with a light-dependent operating procedure, somehow recapitulating a deficit in the interpretation of visual stimuli reminiscent of patients’ CVI [16]. This suggests a specific impairment in the perception and elaboration of visual stimuli in high-order cortices, rather than a generalized defect in learning and cognitive function. Altogether, mouse models point to a scenario in which multiple structures along the visual pathway are impacted by *Nr2f1*-deficiency, from the retina and ON up to the thalamus, the V1 and visual associative cortices together with their connecting tracts.

Interestingly, human *NR2F1* shows similar graded expression along neocortical axes [2,7,21,23], but it is still unknown whether BBSOAS patients display analogous defects in the positioning, size and function of primary and secondary visual areas; functional magnetic resonance imaging (fMRI) analysis and brain tractography could help to tackle this issue in future clinical studies. Undoubtedly, impaired visual learning and aberrant visual processing due to reduced *NR2F1* dosage constitute one of the main features of BBSOAS intellectual deficits, and there is an urgent need of specific early therapeutic interventions to help BBSOAS patients in reaching visual developmental milestones.

## 7. The Many Converging Roads of Intellectual Disability in BBSOAS Patients: From Corpus-Callosum Thinning to Hippocampal and Neocortical Malformations

The second-most prevalent condition in BBSOAS is a moderate to severe ID [38], found in 87% of patients and particularly common upon DBD mutations (94% of patients). Interestingly, even the milder BBSOAS genetic categories, such as patients bearing LBD variants or deletions, show high incidence of ID and speech delay (70% and 80% of described patients, respectively). Despite a high frequency of cognitive symptoms, brain morphological malformations underlying ID in BBSOAS patients are still poorly characterized. Due to the high prevalence of visual deficits in BBSOAS, clinicians tend to focus more on optical features when analyzing MRI brain scans [1,30]. Moreover, the difficulty in obtaining high-resolution MRI scans from young patients also hampers the acquisition of morphological data. Nevertheless, morphological brain malformations have started to emerge, such as a thin corpus callosum (CC) with general presentation or restricted to posterior regions, hippocampal malrotation or dysmorphia, rare white matter loss (demyelination in 14% patients) and cases of localized dysgyria or megalencephaly.

The CC is the largest commissure connecting brain hemispheres, allowing integration between the two cerebral halves. Notably, CC thinning or agenesis could directly contribute to ID, visual problems, motor impairment, speech delay and seizures described in patients [102,103,104]. Consistently, mouse data have demonstrated the role of *Nr2f1* during CC formation as a regulator of the differentiation and migration of late-born cortical neurons [105]. Callosal *Nr2f1*-deficient neurons, found in reduced numbers at postnatal ages, also fail to elongate their axons and topographically innervate the contralateral hemisphere [105,106], ultimately leading to a thinning of the CC, as in human patients. Furthermore, upon fiber elongation via axonal growth, long-range tracts, such as the CC, must be properly myelinated for optimal signal conduction; impaired myelination is associated with human clinical conditions, such as demyelinating diseases or white matter disorders leading to ID, seizures, lack of coordination and other neurocognitive consequences [107]. In mutant mouse models, *Nr2f1* regulates myelination levels in the brain [16,67], similarly to what is observed in BBSOAS patients, suggesting a key role for *Nr2f1* in the maturation of oligodendrocytes and resulting myelination process.

Alongside CC thinning, specific morphological hippocampal defects have been reported in some BBSOAS patients. Since the hippocampus is a key structure for learning, memory and other cognitive processes [108,109,110] and defects in its development can negatively impact such cognitive performances [111,112,113,114], it is reasonable to hypothesize a hippocampal component underlying ID in BBSOAS patients. Mouse experiments support this hypothesis, as *Nr2f1* mutants show defective hippocampal morphogenesis and function [11,12,13]. By regulating both hippocampal progenitor proliferation and neuronal migration during embryonic and early post-natal development [13], *Nr2f1* mainly acts on the development of the dorsal-most hippocampal regions [12], which are involved in spatial navigation, learning and memory [115]. In addition, electrophysiological data revealed impairment of two cellular correlates of learning and memory, long-term potentiation and long-term depression in a *Nr2f1 HET* mouse model [11], suggesting that altered synaptic plasticity may also contribute to BBSOAS intellectual impairment. However, mutant mice also manifest a prolonged fear-memory retention [11], which is in line with unusually good long-term memory reported in up to 75% of BBSOAS patients [38]. This suggests that *NR2F1* might play an important role in the retention of memory in both mouse and human hippocampus. The exact impact of hippocampal abnormalities on BBSOAS patients is still unknown to date, and more studies will be necessary to further assess the contribution of other cortical or subcortical regions.

Recently, a novel cohort of six BBSOAS children showed specific cortical morphological defects, such as polymicrogyria- or dysgyria-like malformations in the supramarginal and angular gyri [8], regions specifically involved in language, vision, spatial cognition, memory retrieval, attention and number processing [116,117,118]. Such malformations were sometimes associated with elongated occipital convolutions, reminiscent of a mild caudal megalencephaly [8]. In mice, *Nr2f1* fine-tunes neural proliferative potential by controlling cell-cycle progression and balancing progenitor pool amplification and local neurogenesis [8,9]. Upon *Nr2f1* loss, a delay in neurogenesis results in amplification of distinct progenitor classes, which ultimately causes caudal cortical expansion [8] and increased overall neocortical volume [11]. The occipital expansion reproduced in mice is somehow reminiscent of the elongated occipital convolutions and megalencephaly reported in some BBSOAS patients, suggesting that common cellular and molecular mechanisms might be shared between mouse and human.

Both the presence of cortical malformations in BBSOAS patients and data coming from mouse models link the *NR2F1* gene to a heterogeneous group of NDDs, called malformations of cortical development (MCDs). MCDs comprise a variable class of morphological abnormalities, such as polymicrogyria and macrocephaly, often associated with ID, autism, speech and motor difficulties and/or epilepsy [119,120,121,122]. Hence, these malformations could underlie some of the key BBSOAS cognitive disorders. However, whether they are a common feature of BBSOAS is still unknown, and a more detailed examination of MRI scans in larger cohorts, with a special focus on occipital and parietal regions, will be needed in future analysis. Furthermore, mouse models could only partially recapitulate these morphological defects [8], as they lack specific human-like features, such as cortical convolutions and abundance of a specific class of neural progenitor termed the basal radial glia, highly present in gyrencephalic species [123,124,125,126,127]. Studies in gyrencephalic mammals, such as the ferret [128], or in human-like systems such as the induced pluripotent stem cell (iPSC)-derived brain organoids [129,130,131], could help unravel *NR2F1* function in mammalian neural proliferation and neocortex gyrification.

## 8. *NR2F1* as an Autism Spectrum Disorder Gene

Autism spectrum disorder (ASD) or autistic traits constitute another major clinical feature of BBSOAS [38]. ASD is a complex group of NDDs characterized by qualitative impairments in social interaction and communication, with repetitive/stereotyped patterns of behavior, interests and activities. Several genomic studies have unveiled a link between disruptive variants in *NR2F1* gene and high susceptibility to ASD phenotypes [26,132,133,134]. As a result, *NR2F1* is now classified as an ASD gene with “suggestive evidence” in the SFARI (Simons Foundation Autism Research Initiative) database [26,31,135]. However, while some BBSOAS patients officially meet the requirements for an ASD diagnosis (38% diagnosed with ASD), some of them have at least autistic features (14.1% patients with autistic features but no official ASD diagnosis) [38], such as repetitive/stereotyped movements in the form of head-banging and hand-flapping, repetitive language, circumscribed interests and self-injurious behaviors, among others [136]. The highly variable behavioral phenotype of BBSOAS also includes attention deficit hyperactivity disorder (ADHD), obsessive compulsive disorder (OCD), pervasive developmental disorder-not otherwise specified (PDD-NOS) and, less commonly, psychosis in the form of auditory hallucinations [42].

Due to the limited number of reported patients, together with the variable expressivity of the BBSOAS behavioral spectrum, it is challenging to distinguish behavioral abnormalities due to ASD or to other clinical features, such as visual impairment [137], epilepsy [138] or ID [139,140,141,142]. As an example, children with visual impairments might display head-banging behaviors [137]. Furthermore, some stereotyped movements in patients with ID and concurrent epilepsy could be the result of unrecognized seizure activity [138], and in general, higher rates of disruptive and self-injurious behavior can be found in patients with combined ASD and epilepsy [143]. Additionally, ID is often associated with stereotyped and aggressive behaviors [143,144]. Finally, obsessive-compulsive behaviors typical of ASD and present in some BBSOAS patients partially overlap with typical symptoms associated with OCD [145]. The most probable scenario for BBSOAS is that multiple cognitive impairments—ASD, ID and epilepsy, among others—could converge on a common (although variable) behavioral spectrum, and overlapping pathologies could synergistically lead to shared phenotypes [146]. Hence, to better discern the different aspects of cognitive impairments characteristic of the BBSOAS behavioral spectrum, further neuropsychological assessments are needed.

As for other BBSOAS features, the use of mouse models has helped to unravel possible etiological mechanisms underlying patients’ autistic features. For instance, some cortical morphological changes reported in *Nr2f1* mutant mice are the result of a regional-specific control of neurogenesis, which, when impaired, might lead to local megalencephaly and/or abnormal gyrification [8,9]. Moreover, morphological changes are also associated with altered neocortical identity upon *Nr2f1* loss, with posterior sensory areas being reduced in size, and a large cortical surface acquiring anterior motor-like properties [7,10]. Interestingly, arealization defects—especially concerning the frontal lobe—regionally altered gyrification of cortical areas and imbalanced neurogenesis leading to early expansion of the brain have all been linked with ASD in patients [147,148,149]. Hence, BBSOAS autistic features and other cognitive impairments could be influenced by impaired arealization upon human *NR2F1* perturbations, but further investigation by fMRI is required to investigate potential areal functional impairments.

Another interesting hypothesis is that ASD and other neurodevelopmental impairments described in BBSOAS patients could derive from an excitatory/inhibitory (E/I) dysregulation due to an imbalance between dorsally generated excitatory glutamatergic neurons and ventrally generated inhibitory interneurons [150,151,152,153,154]. In this respect, a recent report introduced a patient-specific DBD variant in mouse *Nr2f1*, in an attempt to reproduce a genuine point-mutation mouse model of BBSOAS [155]. Mutated *Nr2f1* promoted differentiation of inhibitory neurons, while concomitantly reducing the rate of production of glutamatergic ones, resulting in autistic-like behavioral deficits, such as impaired social interaction. Interestingly, the observed behavioral deficits could be partially alleviated by antagonizing the excessive inhibitory synaptic transmission [155]. Consistently, mutations affecting other ASD risk genes have been shown to trigger a similar E/I imbalance through distinct molecular pathways ultimately converging into shared neurodevelopmental abnormalities [156]. In summary, while the exact molecular and cellular mechanisms leading to autistic phenotypes are still under investigation, we expect *NR2F1* to be soon considered a *bona fide* ASD gene.

## 9. *NR2F1* as a Susceptibility Gene for Infantile Epileptic Disorders

Although not always present (46% of BBSOAS patients), another noteworthy BBSOAS feature is the insurgence of various epileptiform pathologies, comprising infantile and febrile spasms, West syndrome and other forms of epilepsy in children [1,30,38]. Particularly common in DBD patients (53%), epileptic features are more rarely reported in patients with LBD variants or deletions (29% and 27%, respectively), suggesting that epilepsy could represent a BBSOAS feature specifically associated with more severe genetic variants. In some cases, altered electroencephalogram (EEG) patterns—often restricted to the occipital hemispheres—were recorded even in the absence of epileptic seizures [85]. The presence of early-onset epileptic forms in infancy (such as infantile spasms) is of particular importance, as these so-called “catastrophic epilepsies” are associated with poor neurodevelopmental outcome and could inflict additional damage to the developing brain, contributing to ID, memory impairment, attention deficits, developmental delay and autistic features [157,158,159,160,161,162]. The discrimination between brain injuries leading to or caused by seizures has always constituted a great challenge for clinicians [163,164].

The disruption of distinct developmental processes has been hypothesized to underlie the onset of epilepsy [165]. As an example, an imbalance of excitatory pyramidal neurons and inhibitory interneuron subtypes has been proposed as common potential cause of epilepsy and autism [166,167]. Alternatively, interneurons could promote seizures in the initial phase of epileptic activity [168], suggesting that epileptiform events could originate from GABAergic dysfunction *per se*, regardless of their excitatory glutamatergic counterpart. Interestingly, mouse *Nr2f1* is known to regulate the number and type of GABAergic interneurons produced in the ventral telencephalon and reaching the cortex [14,15]. Further, epileptic discharges can also be caused by improper electrophysiological activity of neural circuits due to intrinsic electrical properties of glutamatergic neurons and astrocytes [169,170]. Recent studies show that mouse *Nr2f1* controls the intrinsic electric properties of pyramidal neurons by directly regulating the expression of distinct voltage-gated ion channels as well as the axon initial segment length and diameter [20,171]. By allowing a fine-tuning of action potentials and proper modulation of spontaneous network activity during circuit maturation, mouse *Nr2f1* can ultimately sculpt the emergence of electrical activity in cortical networks [20].

Finally, drug-resistant forms of epilepsy can also be caused by nodular periventricular heterotopia [172,173,174,175], ectopic clusters of grey matter caused by impairment of neuronal migration and connectivity. Defective migration of neuronal cells has been linked to *Nr2f1* loss in mouse [105], possibly due to altered expression of specific cytoskeletal proteins [106]. Consistently, periventricular heterotopia has been reported in some BBSOAS patients alongside seizures [8,29], suggesting a cause–effect link between these two phenomena, whereby clusters of misplaced neurons could act as triggering foci.

The *NR2F1* gene is starting to emerge from unbiased exome-sequencing approaches in individuals with epilepsy [25,176,177,178], and is associated with West syndrome [34], a severe form of infancy epilepsy characterized by clusters of spasms. We propose the inclusion of *NR2F1* in the diagnostic NGS gene panels for epilepsy, which could help families in their diagnostic odyssey to efficient clinical assignment.

## 10. *NR2F1* on the Move: Motor Dysfunction in BBSOAS Patients

The development of the motor system is critical for an individual to experience the environment and engage in social interactions. In NDDs, motor skill impairments are particularly prevalent, to the extent that they are considered by clinicians as one of the first signs of atypical development [179,180,181]. BBSOAS patients have frequently been reported as presenting several motor defects, including delayed motor development, stereotyped and repetitive movements and reduced sensorimotor precision and speed, resulting in defective execution of fine visual motor behavior [33,38]. In children, the typical milestones of motor development are reached at 7 months (sitting), at 10 months (crawling) and around 13 months (walking) [182]. BBSOAS children instead achieve these milestones much later, at an average age of 14, 16 and 33 months, respectively [38]. Interestingly, the incidence of motor delay is higher in patients with DBD variants than those with mutations in other parts of the protein (53% of patients vs. 0% of children bearing all other variants) [38]. Notably, the high severity and penetrance of motor impairments in DBD-variant patients could also be the indirect consequence of other concurring clinical features, such as hypotonia and brain damage following epileptic seizures. Moreover, stereotyped and repetitive movements, such as head-banging and self-injurious behaviors, have been reported in several BBSOAS patients [1,30,38]. The etiological origin of stereotypies is still uncertain, but given the high prevalence of ASD among BBSOAS patients and its association with repetitive behaviors [136], it is possible that these behaviors represent typical ASD features, rather than separate motor symptoms. Another possibility is that the fronto-striatal circuit, responsible for the inhibition of stereotyped repetitive movements, might also be affected in BBSOAS patients, similarly to what is reported for some ASD patients [183].

Abnormal motor behavior can also be observed in the specific context of visual function and eye reflexes. Many BBSOAS patients show nystagmus (repetitive and uncontrolled involuntary movements of the eyeballs), poor object tracking, and saccadic eye movements (rapid, uncontrolled movements of the eyes that abruptly change the point of fixation) [30,38]. In a first thorough characterization of oculomotor skills of a BBSOAS case, the patient showed reduced accuracy when performing specific visual oculomotor tests, compared to age-matched control or ASD individuals [33]. Additionally, Bojanek and colleagues performed a series of tests on general manual skills; the BBSOAS patient showed impaired stopping accuracy and reduced reaction time when compared with both ASD or control age-matched individuals, consistent with a global sensorimotor impairment [33]. This might suggest weakened modulation of the cerebellum on pontine-brainstem burst cells, whereas the slower reaction times could implicate a dysfunction of either the descending cortico-ponto-cerebellar and/or the ascending cerebellar-thalamo-frontal circuits [33]. However, to define the prevalence of these motor dysfunctions among BBSOAS patients and the possible neuroanatomical correlates, a systematic assessment of motor features coupled with MRI examination on bigger cohorts is needed.

In humans, abnormalities of corticopontine and corticospinal descending tracts are usually associated with more broad brain malformations, impacting the execution of voluntary movements and hand dexterity [184,185,186,187,188]. Previous studies in mice helped in describing the development of cortical descending tracts, and noted the existence of several critical steps, each controlled by specific assets of molecular players (revised in [188]). The use of different mouse *Nr2f1* mutants suggested altered wiring in the descending corticopontine and corticospinal tracts, and pointed particularly to the topographic organization of corticopontine projections, an essential process for the control of fine voluntary movements in rodents and humans [189]. Furthermore, the impairment observed in both tangential (areal) and radial (laminar) organization of the neocortical layers upon cortical specific loss of *Nr2f1* function ultimately results in abnormal connectivity between the cortex and its subcerebral targets [7,10,66,190]. This defective connectivity has a deleterious effect on the accuracy of voluntary motor execution and object reaching behavior: although retaining general normal motor capabilities, fine-skilled paw movements show a remarkable impairment in *Nr2f1* mutant animals [10], somehow recapitulating fine motor impairments of BBSOAS patients. Finally, it is reasonable to hypothesize that reduced dexterity could be also impacted by hyperactive features, which have been reported in patients [30,38] and described in *Nr2f1* mouse mutants [191].

Further clinical reports will be necessary to fully characterize motor dysfunctions in BBSOAS patients and to distinguish abnormalities due to ASD from those primarily dependent on other motor deficits. However, independently of their clinical origin, motor dysfunctions might represent a first sign of atypical development in BBSOAS. In fact, early-onset motor impairments often emerge before social and communicative deficits [181,192,193,194]. Hence, they may serve as an early clinical indicator for BBSOAS, as happens for ASD patients [181,192,195].

## 11. Facial Dysmorphia

BBSOAS patients were initially reported not to have a clear *facies typica* associated with their syndromic conditions [1], as dysmorphic facial features could be present but seemed extremely variable and nonspecific. However, with increasing numbers of patient reports, some characteristic features started to emerge, such as ear abnormalities, including prominent ears and bilateral cupped ear helices [1,27,30,35,37], sometimes treated with cosmetic otoplasty. Other facial features include epicanthal folds, small/high nasal bridge, up-slanting palpebral fissures, tall forehead and thin upper lip [8,38]. Finally, dysmorphic facial features additionally included micrognathia and retrognathia [34,37,38], treated in one case with jaw expansion surgery [33]. Data obtained in zebrafish proved Nr2f factors to be essential for cartilage development of the lower jaw [196,197]. Importantly, NR2F1 has been reported to regulate neural crest gene expression and craniofacial morphogenesis by binding to a subset of neural crest enhancers [198], and Nr2f1-mediated BMP2 regulation is key for the differentiation and ossification of human bone marrow stromal cells [199]. These reports point to a role for NR2F1 in the regulation of neural crest development, face cartilage, and more broadly, bone ossification, which could explain not only facial dysmorphic features, but also bone age abnormalities, reported in a BBSOAS patient [35], and short stature [38]. We believe that NR2F1 functions in other tissues, still poorly characterized due to the major focus being on brain development, could have a great impact on non-neural tissue, and explain specific BBSOAS clinical features involving cartilage and bone development.

## 12. Feeding Problems and Mouth Stuffing, a Common but Poorly Characterized BBSOAS Feature

A common congenital feature of BBSOAS patients is oromotor dysfunction, consisting in poor swallowing, sucking and chewing, ultimately leading to feeding abnormalities and in some cases requiring gastric tube feeding. Notably, poor feeding could converge on other clinical features by contributing to the global developmental delay and short stature observed in young patients [200,201]. BBSOAS feeding deficits could be caused and/or worsened by hypotonia [202,203,204], one of the most common symptoms in BBSOAS patients. However, studies on mice open an interesting new interpretation of BBSOAS feeding problems. In fact, *Nr2f1 null* mouse mutants die perinatally of dehydration and starvation, due to multiple defects in cranial glossopharyngeal ganglion (IX) and nerve formation [205]. The ninth nerve supplies both sensory and motor innervation to the pharynx and root of the tongue, innervates the soft palate and induces salivary secretion essential for swallowing by innervating the parotid gland. Hence, even if BBSOAS feeding deficits may be caused indirectly by reduced muscle tone early in infancy, it is tempting to speculate that they could directly result from morphological defects of the IX cranial nerve as found in mouse models.

## 13. The Emerging BBSOAS Genotype–Phenotype Correlation from a Molecular Point of View: NR2F1 Dimers and the Dominant-Negative Effect

As described in earlier sections, BBSOAS patients are characterized by multiple clinical features. Such high clinical heterogeneity could depend—at least in part—on an equally high diversity in terms of *NR2F1* genetic variants. While experimental models are helping to elucidate possible cellular alterations leading to some symptoms, a careful evaluation of NR2F1 function at a molecular level is also necessary to uncover potential genotype—phenotype correlations and obtain a deeper comprehension of BBSOAS features.

The first major genetic difference among BBSOAS patients consists of whether the function of one *NR2F1* allele is lost by whole-gene deletion or by point mutations. On one side, BBSOAS patients with chromosome deletions physically lose one copy of the *NR2F1* gene, resulting in reduced protein dosage and functional haploinsufficiency. Conversely, patients carrying deleterious (loss-of-function) missense mutations in one allele still express mutated NR2F1 forms. Missense point variants are mainly reported in the coding sequence of two functional domains highly conserved across members of the nuclear receptor family: the DNA-binding domain (DBD) consisting of two zinc-finger domains, and the ligand-binding domain (LBD) (Figure 1). The structure of the NR2F1 LBD and the identity of candidate chemical agonists fitting the ligand-binding pocket start to be characterized [206,207]; however, as the physiological ligands remain unknown, NR2F1 is still ascribed as an orphan nuclear receptor. The LDB is also responsible for protein dimerization, as NR2F monomers combine among them or with other members of the same family (forming homo- or heterodimers, respectively) to regulate gene expression [3,66]. Similarly to other nuclear receptors, NR2Fs also display two activating function domains (AF-1 and AF-2), necessary for binding co-factors that can be either activated by conformational changes induced by ligand binding [208,209,210], or in some cases, be constitutively active [211].

Since NR2F1 binds the DNA in the form of homodimers or heterodimers [4,212], point-mutation variants could result in complex dominant-negative effects, in which the mutated form competes with the *wild-type* protein (or with other nuclear receptors) for dimer formation. Indeed, a dominant-negative effect of DBD mutations was proven by in vitro luciferase assays [30,155]. Such a molecular mechanism would explain why BBSOAS patients with point mutations falling in the DBD often have a high prevalence and overall increased clinical severity, such as greater severity of speech and motor disabilities, compared to patients with other variants [38]. Additionally, epileptic features are more often reported in DBD-mutated patients (74% vs. 37% in patients with other variants, as previously reported) [38]. However, due to the low number of BBSOAS patients described to date, little is yet known about the emerging genotype–phenotype correlation and the molecular mechanisms underlying such correlations.

The 92 patients reported by different studies (Table 1) show a diverse array of deletions or point variants, completely ablating one *NR2F1* allele or affecting its sequence, respectively. We provide here a classification of the *NR2F1* perturbations observed so far and speculate on possible genetic/molecular mechanisms that could impact dimer formation and NR2F1 functioning (Figure 4), possibly explaining the molecular origins of the genotype–phenotype correlation. For this purpose, we correlated each patient group to a severity index based on the presence or absence of the main BBSOAS clinical symptoms (abnormal MRI, DD, ID, visual system defects, epilepsy, ASD or autistic-like traits and hypotonia) and ranging from 1 (one symptom only) to 7 (when all symptoms are present). The severity index and the prevalence of each clinical category in different genetic groups are reported in Table 5.


**BBSOAS genetic categories and *NR2F1* function:**
**Whole-gene deletions or small indels. (Severity index: 4.33).** A few BBSOAS patients so far have been reported with chromosome deletions spanning from 582 Kb to 5 Mb in size, all resulting in whole-gene ablation and complete loss of one *NR2F1* allele [1,27,28,29,30,38]. The likely consequence of such deletion is a halved NR2F1 protein production, i.e., haploinsufficiency, as was proven by protein quantification in skin fibroblast extracts [30]. However, it must be noted that large deletions and complex chromosomal aberrations can involve additional genes located next to the *NR2F1* locus, such as *FAM172A*, *POU5F2*, *MIR2277* and *lnc-NR2F1*, adding to the complexity of the condition, and possibly leading to supplementary congenital abnormalities, such as periventricular heterotopia and deafness, among others [27,28,29]. Interestingly, specific deletions only affecting the region adjacent to the *NR2F1* gene, where the long non-coding (*lnc*)*-NR2F1 RNA* is located, were also shown to cause neurodevelopmental conditions with developmental and speech delay [213], probably due to the ability of *lnc-NR2F1* to control the expression of autism-associated neural genes in a similar way to its corresponding protein-coding gene. These data suggest that (i) multiple genes could synergistically converge to cause similar neurological conditions, and that (ii) the loss of other genes together with *NR2F1* in BBSOAS patients with large deletions could exacerbate the clinical features and further increase their heterogeneity.**Translation initiation variants (TIVs)****. (Severity index: 5.33).** BBSOAS patients with missense variants falling in the translation initiation codon (ATG) show decreased NR2F1 protein resulting from reduced efficiency of both translational and transcriptional processes [30]. Notably, the third codon of the *NR2F1* gene sequence is also an ATG, raising the possibility that an alternative initiation site could be present and potentially compensate for the loss or mutation of the main one. However, diminished NR2F1 levels measured in cells from these patients suggest that the second ATG site is not able to efficiently serve as an alternative start codon [30]. The production of a half dosage of the functional NR2F1 protein, without any mutated form competing for dimer formation, could make these patients more similar to BBSOAS individuals with whole-gene deletions than to patients carrying missense DBD/LBD variants, hence leading to milder phenotypes. However, this has been recently questioned by the report of a severe BBSOAS case owing to an *NR2F1* start codon variant [85]. While the severity index shows an intermediate severity compared to the two groups, statistical analysis on larger cohorts will be necessary to better detail the phenotypes associated with TIVs.**DBD missense variants (or DBD in-frame deletions)****. (Severity index: 5.62).** The effect of missense mutations is tightly linked to the structural and functional importance of the affected amino acids and the protein region in which they are located. Based upon evolutionary conservation of distinct aminoacidic positions, bioinformatic prediction software can be used to identify highly conserved regions, and to evaluate whether they are intolerant towards variation and therefore potentially pathogenic [214,215,216]. BBSOAS pathogenic point mutations falling in the DBD and impacting the 3D structure of evolutionarily conserved sites in the zinc-finger domains have strong effects on NR2F1 structure and function, and consequently, on the clinical phenotype. As an example, substitution of a highly conserved zinc-finger motif of the DBD of NR2F1 leads to heavy changes in molecular structure and stability, as predicted in silico [36]. By contrast, missense variants located in less conserved regions adjacent to the DBD showed only a reduced transcriptional activity in the luciferase assay, in contrast to the almost abolished activity of DBD variants falling in crucial zinc-finger or structural sites [30]. As discussed above, the high penetrance of BBSOAS features following DBD missense mutations, compared to the complete loss of one allele by whole-gene deletion, could result from a possible dominant-negative effect of mutated NR2F1. Indeed, by forming non-functional dimers with the *wild-type* NR2F1 (produced by the normal allele), mutated NR2F1 proteins could ultimately affect 75% of the total pool of NR2F1 dimers in the cell (Figure 4). Heterodimer formation with other nuclear receptors (NRs) could be affected too, presumably resulting in a 50% loss of functional NR2F1-NRs heterodimers (Figure 4). The high number of reported DBD missense mutations, associated with the highest severity index among BBSOAS genetic groups, confirms the functional relevance of this region.**LBD missense variants****. (Severity index: 3.76).** Patients with variants in the LBD manifest milder developmental delay, often lacking hypotonia, speech defects, seizures and repetitive behaviors [35], consistent with the lowest severity index among BBSOAS genetic categories. As the LBD is necessary for NR2F1 dimerization, it is reasonable to think that mutations falling on interacting surfaces could hamper dimer formation. While this could decrease NR2F1 activity in general, such a situation could also affect the formation of dimers between normal and mutated NR2F1, hence resembling whole-gene deletions, with no (or lower) dominant-negative effect. The LBD also contains the Activating Function domain 2 (AF-2), which is necessary for co-factor binding. While the exact nature of NR2F1 protein partners in neural cells remains elusive, the loss of such interactions following LBD mutations could be responsible for some BBSOAS features. Further studies will be necessary to dissect the specific function of distinct domains in the LBD and the impact of distinct mutations on dimer formation, as well as the identity of NR2F1 physiological co-factors. This will lead to further insights into NR2F1 molecular functions and how they correlate with patients’ symptoms.**Truncation or frameshift followed by truncation variants****. (Severity index: 5.18 and 5.29, respectively).** Nonsense mutations result in premature termination of the protein (i.e., truncation of the peptide sequence) and can have profound impacts on gene function [217]. Obviously, the portion of the protein that is lost depends on the precise location of the aberrant termination site. The low number of described patients carrying NR2F1 truncations makes it challenging to gain a clear picture of the genotype–phenotype correlation for those pathological cases. However, it is reasonable to think that early truncations would trigger nonsense-mediated decay of the *mRNA*, resulting in loss-of-function, whereas late truncation variants that escape nonsense-mediated decay may create an abnormal, but stable, truncated protein, causing dominant-negative or neomorphic effects [218]. Hence, early variants leading to haploinsufficiency would be more similar to whole-gene deletions and may also protect against more detrimental phenotypic effects (i.e., dominant-negative effect), presumably leading to milder phenotypes [218,219]. Consistently, a recently reported patient presenting NR2F1 early truncation showed a mild phenotype [40], whereby one case has high-functioning ASD with superior verbal abilities [33]. By contrast, late variants that do not affect the dimerization region of the LBD could show a dominant-negative effect, similarly to what is reported for point variants, and ultimately leading to highly penetrant symptoms. Taking all truncation variants together, their collective severity index is more severe than that of deletions, similarly to variants affecting the starting codon.


A further complication is represented by frameshift variants, which can determine large-scale deleterious changes to the overall polypeptide length and chemical composition. Moreover, they are often followed by truncation at a termination site located at variable distance. By dramatically altering the protein sequence, truncating variants with frameshift can result in heavy conformational changes and can have much stronger effects on disease risk [220]. The acquisition of toxic properties can constitute a deleterious event following frameshift [218], further impacting cellular physiology and biochemical processes, thus being more deleterious than the loss of the gene itself. Consistently, a patient with frameshift variant and truncation in the NR2F1 LBD has been reported to show a severe phenotype, including epilepsy [39]. Their severity index shows that these variants could be slightly more severe compared to simple truncations. In vitro assays and in silico predictions of 3D protein structure, together with accurate description of a higher number of patients, could help unravel the exact effect of NR2F1 truncations falling in different protein regions and the possible pathogenic impact due to changes in the aminoacidic sequence following frameshift.

In summary, the severity of BBSOAS symptoms could be the result of both *NR2F1* haploinsufficiency and dominant-negative effects affecting protein availability on one side and dimer formation on the other. Consistent with previously reported data [30,38], the severity index presented here points to a scenario where a dominant-negative effect of DBD genetic variants results in the most severe clinical outcomes. Indeed, DBD variants lead to high percentages of main BBSOAS symptoms (Table 5), such as DD (90.6% of patients), ID and/or speech delay (93.8%), CVI (53.1%), OA (78.1%), seizures (62.5%) and hypotonia (75%). On the contrary, the milder phenotypes would be associated with LBD pathogenic variants, showing a decreased prevalence of distinct symptoms when compared to other genetic groups, notably concerning DD (70.6%), ID (70.6%), OA (47.1%), hypotonia (35.3%) and epilepsy (29.4%). It is interesting to note that deletion variants, with respective prevalence of DD (93.3%), hypotonia (60%) and optic disc malformations (33.3%), manifest a higher severity index than LBD point mutations (4.33 vs. 3.82). Similarly, translation initiation variants (TIVs), previously compared to gene deletion in terms of symptom severity, display a quite high severity index of 5.22, due to high prevalence of DD, ID, optic atrophy and very high incidence of hypotonia (88.9%). We expect that the continuous report of novel patients will gradually refine the calculations of clinical prevalence, allowing a better understanding of symptom severity in the distinct genetic groups. It is intriguing to notice that truncation and frameshift/truncation variants show a particularly high prevalence of CC malformations (63.6% and 42.9%, respectively), OA (72.7% and 71.4%, respectively) and DD (90.9% and 100%, respectively). Furthermore, frameshift/truncation variants have a high prevalence of ONH (57.1%) and high risk of epilepsy (57.1%), ASD (71.4%) and hypotonia (57.1%). Altogether, possible pathogenic effects due to frameshift variants cannot be excluded at this point. However, the number of reported patients carrying these types of variants is very limited and could affect the calculation of symptom prevalence.

Overall, the high clinical heterogeneity of BBSOAS patients could be a direct consequence of a yet not well characterized genotype–phenotype correlation, in which specific *NR2F1* variants trigger respective clinical outcomes. Careful and consistent clinical investigation of newly reported BBSOAS patients would be a key factor to improve the comprehension of BBSOAS clinical features and genotype–phenotype correlation. In parallel, a more comprehensive characterization of NR2F1 molecular function in both physiological and pathological conditions will help predicting the full spectrum of effects of distinct genetic conditions at a cellular and molecular level and will allow the inference of possible consequences in terms of BBSOAS symptoms.

## 14. Discussion

The continuous reporting of newly identified BBSOAS patients, made possible by the effort of their families and clinicians, is rapidly expanding our knowledge of this rare autosomal dominant syndrome. However, the variability of clinical descriptions, depending on the scope and depth of the respective study, make it challenging to compare clinical BBSOAS reports from different sources. Before the discovery and naming of BBSOAS as a unique syndrome, individuals with *NR2F1* mutations were included in autistic, epileptic or other clinical groups, depending on their most prevalent features, leading to poor description of other, apparently unrelated, symptoms. Studies on larger cohorts of patients with *NR2F1* pathogenic variants would further define BBSOAS as a syndromic condition and would lead to a more complete characterization of clinical features [30,38]. Despite this, it remains challenging to compare clinical descriptions from different studies. As an example, ophthalmological features often include OCT scans of the retina and eye fundoscopy, but fewer visual acuity measurements, and an ophthalmological analysis was performed on a large group of patients only recently [41]. Furthermore, MRI analysis of some BBSOAS patients reported morphological malformations of the neocortex [8], but such analysis is still lacking in the vast majority of patients. To help future investigations, it would be necessary to compile a standardized list of clinical exams to regularly perform on newly diagnosed BBSOAS patients. Focus should be on both the clinical challenges BBSOAS-diagnosed children could face (see Table 2) and the clinical evaluations to be implemented (see Table 3).

In light of the rarity of this syndrome, the best approach to date would be to use uniform criteria and to perform in-depth clinical research studies on a higher number of patients, taking advantage of family conferences bringing BBSOAS patients together, as recently undertaken during a meeting organized by the USA parent association [38]. The increase in the number of diagnosed patients would allow exploration of the impact of distinct *NR2F1* genetic perturbations at a cellular and molecular level, and of the ways this contributes to their pathophysiological heterogeneity. Furthermore, a unified registry for BBSOAS patients would facilitate the clinical research on both already reported and newly identified patients. The USA NR2F1 Foundation is moving in this direction (*BBSOAS Patient Registry*, under preparation via Across Healthcare Matrix Platform). Further help could also come from public databases reporting published *NR2F1* variants, using standard nomenclature to describe both molecular and phenotypic anomalies. Tools such as the LOVD database will help to further refine the BBSOAS clinical synopsis and allow for a better comparison of patient symptoms [32].

A further challenge in the analysis of BBSOAS patients comes from the complex syndromic nature of their condition. NDDs often show comorbidity of several clinical features, such as ASD, developmental delay, ID and epilepsy [166,221], and BBSOAS makes no exception. Such recurrent comorbidity sustains the idea that the impairment of common mechanisms orchestrating brain development and function could converge on similar pathophysiological phenotypes via shared cellular and molecular pathways [222]. In such a scenario, the crosstalk between different disorders makes it extremely difficult to understand the temporal order of appearance of distinct clinical features and the causative links among them. In the specific case of BBSOAS, for example, ID could result from multiple factors: visual impairments, seizures, cortical or hippocampal malformations, among others. Visual impairments, on the other side, could be further impacted by mitochondrial dysfunction, reported in two patients [42,43], as ON axonal damage easily results from energetic failure secondary to impairments of the mitochondrial respiratory chain [45]. Furthermore, we cannot exclude that both ID and CVI result from secondary damage caused by severe epileptic infantile spasms, as this catastrophic infant epilepsy has been shown to heavily damage visual system integrity and brain development in general [161,162,223,224,225,226].

An example of complex comorbidity in a BBSOAS-like context comes from an *Nr2f1*-haploinsufficient mouse model recapitulating visual syndromic features, which showed concomitant RGC decreased survival, oligodendrocyte hypomyelination and astrocyte inflammation in the ON of developing mutants [16,41]. In such a complex scenario, understanding the specific contribution of each distinct player in disrupting the ON physiology is most probably unfeasible. A possible solution would be to employ conditional mouse models, in which the use of cell- and time-specific *Cre*-*recombinases* allows the removal of *Nr2f1* in a very targeted way, thus allowing dissection of its role in specific contexts [5]. As an example, conditional *Nr2f1 knock-out* in the oligodendrocyte lineage could help in discerning the impact of oligodendrocyte dysfunction and/or astrocyte inflammation in the ONH patient phenotype. In summary, while *Nr2f1* constitutive mouse models better recapitulate the BBSOAS condition in toto, *Nr2f1* conditional ones help dissect the causative and temporal link between distinct pathological features (reviewed in [5]).

The presence of comorbidities and the phenotypic clinical overlap with other complex syndromic conditions has led to cases of misdiagnosis in BBSOAS patients [44]. This can be avoided by applying an unbiased sequencing approach, such as whole-genome sequencing (WES), to patients with undiagnosed NDDs. WES successfully identified BBSOAS patients among individuals with previously unsolved/unspecific forms of ID [227,228] or epilepsy [229], and the comparison of NGS data helped in compiling the annotation of numerous additional *NR2F1* variants in online repositories [230], comprising VUS (variants of unknown significance), likely benign variants and likely pathogenic variants. In this sense, WES or other whole-genome-sequencing approaches should be used as a first choice in the clinical armamentarium, rather than as a last-resource genetic test [231,232,233]. This approach would not only favor an early and accurate molecular diagnosis, but also constitute the foundation of rational and effective therapeutic choices, leading to significant changes in the clinical management of previously unsolved clinical cases [234,235]. However, some NDDs can also be caused by de novo somatic mutations, which occur post-zygotically and are thus present in only a subset of the cells/organs of an affected individual [236]. Therefore, additional BBSOAS patients with somatic mutations in brain tissue only could remain unreported after WES sequencing on blood samples.

Another open question is whether BBSOAS syndromic traits are completely dependent on genetic factors (i.e., *NR2F1* perturbations) or can also be influenced by the environment, a plausible mechanism for NDDs. As an example, interactions between genetic and environmental factors have been proposed to explain the complex physiopathology of ASD [237,238,239,240]. The concomitant presence of environmental and genetic influences could further explain the high variability in terms of presence/absence and severity degree of BBSOAS features. A recent report described highly similar, almost overlapping symptoms in two BBSOAS monozygotic twins [36], suggesting a strong genetic component for BBSOAS. However, it would be interesting in the future to compare the clinical descriptions of patients with different genetic backgrounds (i.e., non-consanguineous) and from different familial environments but bearing identical missense mutations. This would allow the gathering of new insights into a putative interplay between genetic and environmental components. However, due to discrepancies in patient descriptions among distinct cohorts discussed above, this will be a challenging task.

Regarding cellular and molecular levels, most of the data coming from classical studies on NR2F1/Nr2f1 molecular function and dimer formation have been obtained in non-physiological conditions or in non-neural cells. Moreover, an in silico prediction of the NR2F1 3D protein structure remains partially elusive to date, with the DBD being the only domain to have been resolved as yet (PDB entry: 2EBL). To overcome the lack of NR2F1 LBD structure, and thanks to the high similarity with its homolog NR2F2 [241], the 3D structure of the latter is currently used as an alternative (PDB entry: 3CJW); however, to model the LBD in its active conformational state, the RXR LBD structure is preferred (PDB entry: 1FM6) [206] even though it only shows a 40% homology to the actual NR2F1 sequence. Further, the exact 3D structure of the linking hinge between LBD and DBD is still missing, as it is too variable to be easily predicted. Further studies unraveling, on one side, the 3D structure of NR2F1, and on the other side, its function in a more reliable and physiological context (i.e., living human neural cells), are of extreme importance, as they would allow the prediction of changes in protein–protein and protein–DNA interactions following specific *NR2F1* pathogenic mutations.

Finally, a tight link between clinical reports and experimental research is fundamental for advancing our understanding of BBSOAS and for planning future treatments. One of the key questions to be addressed through research is to what degree BBSOAS is due to altered neurodevelopment vs. altered neuronal homeostasis in the postnatal individual. The greater the contribution of ongoing processes, the greater the potential for therapeutic intervention. As more BBSOAS patients are being reported, and as clinical features are better described, the time will come to test possible therapeutic approaches aiming at improving some patients’ symptoms. As an example, studies in mice showed the feasibility of chemical drug treatments to restore correct myelination levels in the context of several acute or syndromic pathologies, as well as in the ONs of a BBSOAS-like model [16,68,69,70,71,72,75,242,243]. It could be interesting to screen for efficient remyelinating drugs on patient-derived 3D brain organoids containing oligodendrocytes or 2D-differentiated oligodendrocyte precursors [244,245,246,247,248], to identify potential treatments for BBSOAS myelination defects, and more generally, to employ an in vitro BBSOAS-like set-up as a tool for the screening of novel myelinating chemical compounds. Finally, for individuals with point mutations in the DBD, one could envision a therapeutic approach with antisense oligonucleotides targeting the mutated version of *NR2F1*. Although not being expected to cure BBSOAS, it might lead to an amelioration of the phenotype, shifting it into the milder range, such as what is seen among individuals with heterozygous deletions.

In conclusion, the great breadth and variability of BBSOAS symptoms, due to distinct forms of *NR2F1* deficiency, could result from the disruption of converging pathways regulating brain development. The clinical description of newly reported BBSOAS patients, in combination with experimental research in vitro and in vivo, will ultimately allow for a better comprehension of this recently described syndrome and the roles played by *NR2F1* during early development.

## Figures and Tables

**Figure 1 cells-11-01260-f001:**
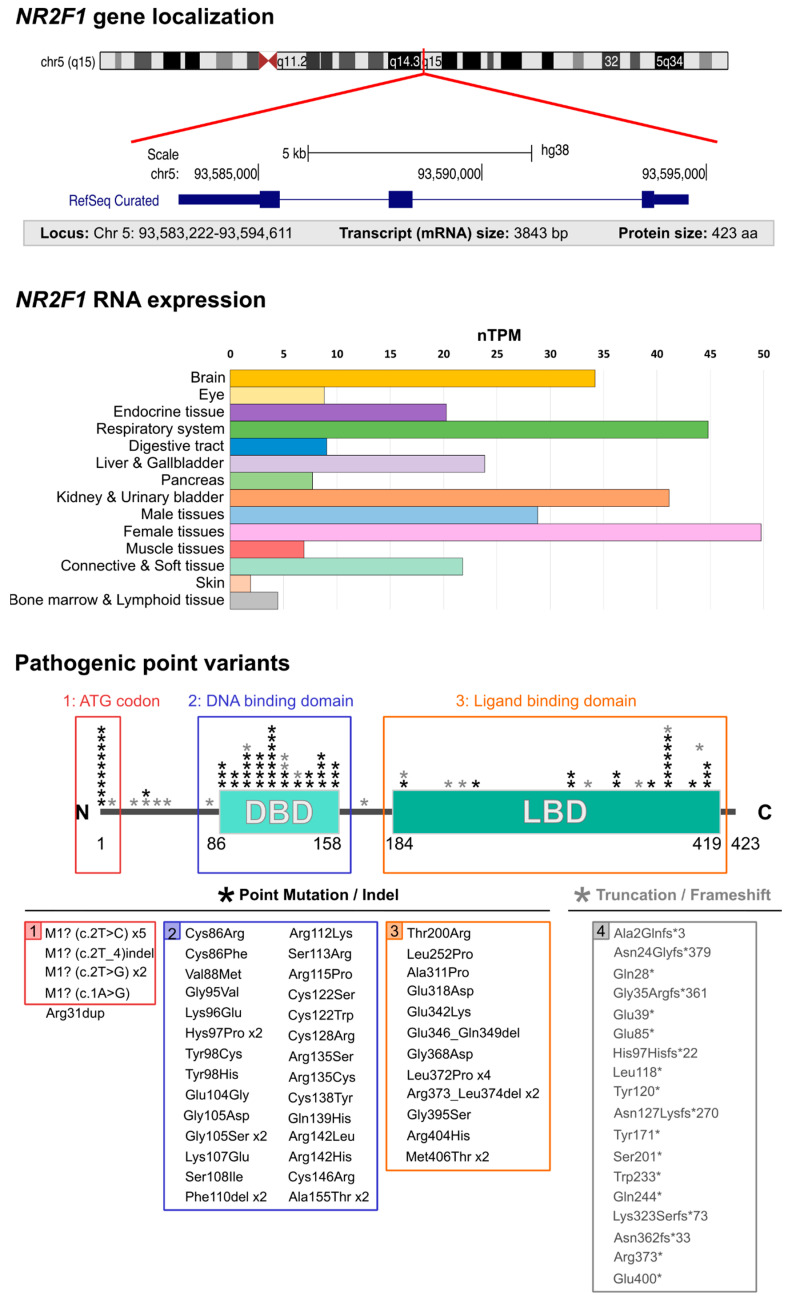
**Schematic representations of *NR2F1* gene localization, expression profile and pathogenic point variants.** The human *NR2F1* gene, located on chromosome 5 (region 5q14–q15), codes for a 3.824 base pairs (bp)-long transcript containing three distinct exons, translated into a 423 amino acid (aa)-long protein (Source: Human hg38 chr5:93583222-93594611 UCSC Genome Browser v427). The *NR2F1* expression profile in different tissues and organs is shown as normalized transcript per million (nTPM). Source: human transcriptome dataset at Human Protein Atlas (HPA) (Query: ENSG00000175745-NR2F1). *NR2F1* haploinsufficiency in BBOSAS patients is caused by gene deletion or by loss-of-function mutations affecting one allele. Small indels and point variants (black asterisks) tend to fall in the ATG starting codon (1), in the DBD (2) or in the LBD (3). Protein truncations (or frameshift variants followed by truncation at variable distance) are listed with a grey asterisk (4). All variants indicated by asterisks are also listed in the boxes, grouped by gene region or variant type. Whole-gene deletions, not shown here, are listed together with point variants in Table 1.

**Figure 2 cells-11-01260-f002:**
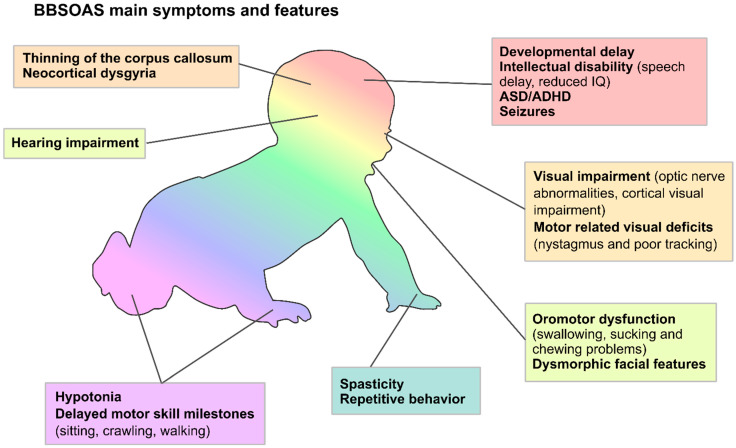
**BBSOAS clinical spectrum**. List of main BBSOAS symptoms and features, subdivided by affected systems. While developmental delay, intellectual disability and optic atrophy are the most common features (88%, 85.9% and 66.3% of patients, respectively), other symptoms are less common, such as CVI (44.6%), epilepsy (46.7%), ASD or autistic traits (39.1% and 14.1%, respectively), hearing impairment (11%) and hypotonia (62%). Abbreviations: ADHD, attention deficit hyperactivity disorder; ASD, autism spectrum disorder; CVI, cortical visual impairment; IQ, intelligence quotient.

**Figure 3 cells-11-01260-f003:**
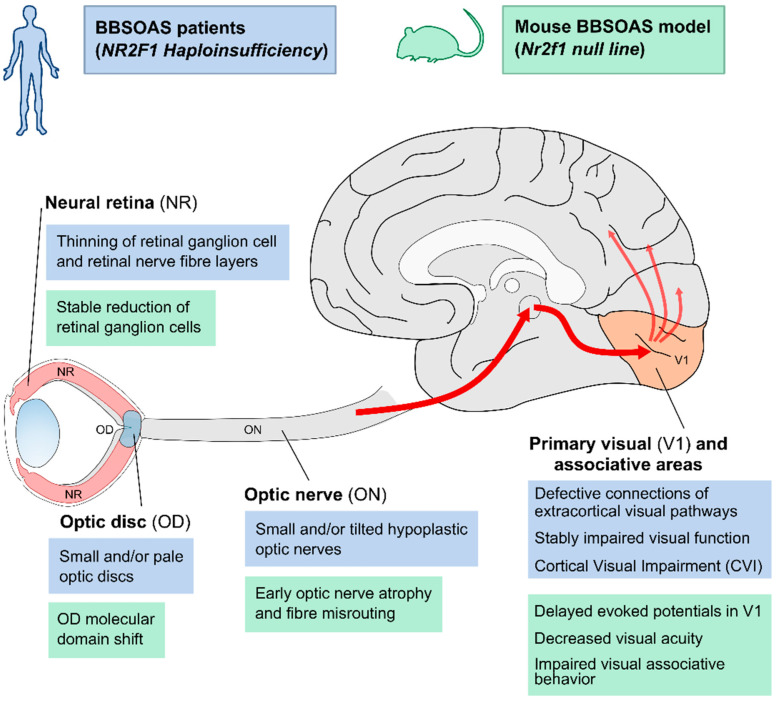
**Overview of structural and functional defects along the visual pathway in BBSOAS patients and corresponding *Nr2f1* mutant mouse models.** CVI and other visual impairments reported in BBSOAS patients (blue boxes) might build upon structural impairment affecting several structures in the visual system, such as the neural retina (NR), the optic disc (OD), the optic nerve (ON), the primary visual area of the neocortex (V1) and its connections to secondary associative areas. The use of *Nr2f1* mutant mouse models (green boxes) have helped in elucidating the molecular, cellular and functional mechanisms that could potentially cause the defects observed in patients.

**Figure 4 cells-11-01260-f004:**
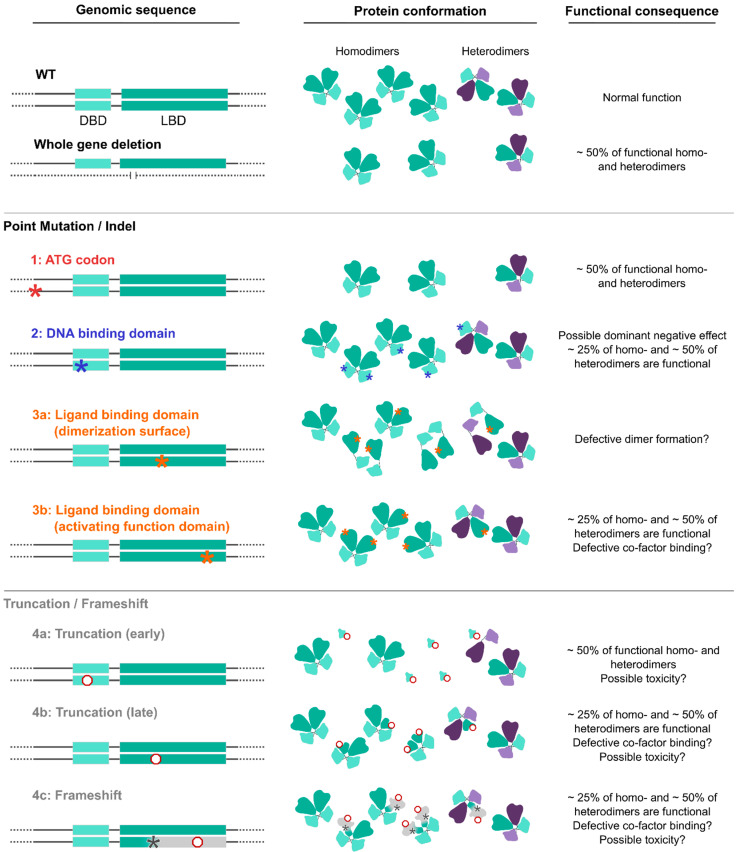
**NR2F1 molecular functioning and dimer formation hypothetical predictions upon distinct genetic perturbations**. Distinct deletions and mutations affecting *NR2F1* genomic sequence (left column) could result in specific scenarios of impaired quantity and/or quality of NR2F1 homo- and heterodimers (central and right columns), possibly explaining a genotype–phenotype correlation. While decreased *NR2F1* expression has been proven for some ATG point variants and for gene deletions, other scenarios—such as the dominant-negative effect of DBD- or LBD-mutated variants—are poorly understood to date. In a similar way, the possible consequences of the production of truncated NR2F1 forms are purely theoretical and will need further experimental assays to be confirmed and characterized. The exact 3D structure of nuclear receptor dimers and their formation rate, the identity of NR2F1 cofactors and the possible toxic effect of frameshift/truncated variants will require further studies. The asterisks indicate point mutations (color-coded following their genetic category), whereas stop symbols (in red and white) represent truncation sites.

**Table 1 cells-11-01260-t001:** **Updated list of *NR2F1* variants and clinical description of BBSOAS reported patients.** BBSOAS patients, identified by their protein variant and—when available—by their LOVD identifier, are listed following the chronological order of reports and publications describing their cases. Main clinical signs include altered brain morphology as observed by MRI, developmental delay (DD), intellectual disability (ID), visual system deficits, early-onset epilepsy and seizures (EOE/S), autism spectrum disorder (ASD) and behavioral abnormalities and hypotonia. The severity index of each patient has been calculated based on the presence/absence of the main clinical signs, ranging from 1 (one symptom category only found in the patient) to a maximum of 7 (all listed clinical signs are present, to different extent, in the same patient). An extended version of these data, with additional columns describing other less common clinical features, is available in Appendix A. *List of references*: AK13, Al-Kateb et al., 2013; BA19, Balciuniene et al., 2019; BE18, Bertacchi et al., 2018; BE20, Bertacchi et al., 2020; BR09, Brown et al., 2009; CA09, Cardoso et al., 2009; BO14, Bosch et al., 2014; BO20, Bojanek et al., 2020; CH16, Chen et al., 2016; DI16, Dimassi et al., 2016; EL17, Eldomery et al., 2017; GA21, Gazdagh et al., 2021; HF15, Hino-Fukuyo et al., 2015; HF17, Hino-Fukuyo et al., 2017; HO20, Hobbs et al., 2020; JS20, Jezela-Stanek et al., 2020; JU21, Jurkute, Bertacchi et al., 2021; KA17, Kaiwar et al., 2017; MH18, Martín-Hernández et al., 2018; MI14, Michaud et al., 2014; MI20, Mio et al., 2020; PA19, Park et al., 2019; SA13, Sanders et al., 2013; ST20, Starosta et al., 2020; RE20, Rech et al., 2020; RO20, Rochtus et al., 2020; VI17, Vissers et al., 2017; WA20 Walsh et al., 2020; ZO20, Zou et al., 2020. *Abbreviations*: CB, cerebellum; CC, corpus callosum; CS, corticospinal tract; D, deletion; DBD, DNA binding domain; DD, developmental delay; DM, delayed myelination; DMD, delayed motor development/poor coordination; DQ, developmental quotient; EOE/S, early-onset epilepsy/seizures; FI/NFI, frameshifting indel/non frameshifting indel; FS, febrile seizures; GCL, ganglionic cell layer; GVI, general visual impairments; HP, hippocampus; HPM, hippocampal malrotation; ID, intellectual disability; IQ, intelligence quotient; IS, infantile spasms; LBD, ligand binding domain; LV, lateral ventricle; LVA, low visual acuity; MCP, macrocephaly; MD, microdeletion; MM, missense mutation; OA, optic atrophy; OC, optic chiasm; OCB/RB, obsessive-compulsive/repetitive behaviors; OD, optic disc; ON, optic nerve; ONH, ON hypoplasia; P/SOD, pale/small optic disc; PDD-NOS, pervasive developmental disorder—not otherwise specified; PH, bilateral periventricular heterotopia; RB, repetitive behavior; RNFL, retinal nerve fiber layer; TIV, translation initiation variants; WGD, whole-gene deletion; WM, white matter. For the extended report, please refer to Appendix A.

References	LOVD Database ID; Patient ID	Variant Type	Variant (Protein)	MRI (General; Optic Nerve and Cortical Morphology)	DD	ID	Visual System Defect(s) and Visual Deficit	EOE/S	ASD Behavioral Abnormalities	Hypotonia	Severity Index
BR09	BE18, #1	De novo deletion (400–500 Kb MD at breakpoints following paracentric inversion)	Deleted	Cranial nerve abnormalities	Yes	ND	ND	ND	ND	Yes	3
CA09	BE18, #2	De novo deletion	Deleted	PH	Yes	Yes; speech delay	Coloboma	FS	ND	Yes	6
CA09	BE18, #3	De novo deletion	Deleted	PH	Yes	Yes; speech delay	ND	IS	ND	ND	4
CA09	BE18, #4	De novo deletion	Deleted	PH; HPM; MCP; polymicrogyria	Yes	Yes; speech delay	ND	Yes	ND	Yes	5
AK13	BE18, #5; RE20, #27	De novo deletion (582 Kb)	Deleted; (del. includes FLJ42709, FAM172A, POU5F2, and MIR2277)	OA (small OC)	Yes, DMD	No but speech delay	OA; GVI	No	ADHD	Yes	6
SA13	BE18, #6; RE20, #53	De novo MM in LBD	p.Arg404His	ND	ND	ND	ND	ND	ASD	ND	1
BO14	LOVD: NR2F1_000001; BO14, #2; BE18, #8; RE20, #13	De novo MM in DBD	p.Ser113Arg	OA (small OD and OC)	Yes	No or ND	OA; P/SOD; CVI; GVI	ND	ND	Yes	4
BO14	LOVD: NR2F1_000002; BO14, #1; BE18, #7; RE20, #14	De novo MM in DBD	p.Arg115Pro	normal	No	Yes (IQ 48)	OA; P/SOD; small ON; CVI; GVI	ND	ND	ND	2
BO14	LOVD: NR2F1_000003; BO14, #3; BE18, #9; RE20, #48	De novo MM in LBD	p.Leu252Pro	ND	Yes	Yes (IQ 55–65)	P/SOD; CVI; GVI	ND	ND	Yes	4
BO14	BO14, #4; BE18, #10; RE20, #28	Deletion (0.83 Mb)	Deleted; (del. includes FAM172A, KIAA0825)	ND	No	Mild (IQ 61–74)	P/SOD; CVI; GVI	ND	ND	ND	2
BO14	BO14, #5; BE18, #11; RE20, #24	De novo deletion (2.83 Mb)	Deleted; (del. includes FAM172A, KIAA0825, ANKRD31)	Normal or ND	Yes	No (IQ ND)	P/SOD; small ON; CVI; GVI	ND	ND	ND	2
BO14	LOVD: NR2F1_000004; BO14, #6; CH16, #12; RE20, #12	De novo MM in DBD	p.Arg112Lys	Normal or ND	Yes	Yes (IQ 52)	OA; P/SOD; mild GVI	ND	ASD; OCD	ND	4
HF15; HF17	LOVD: NR2F1_000018; CH16, #14; RE20, #18	De novo MM in DBD	p.Arg135Cys	Normal or ND	Yes, DMD	Yes (DQ < 20); speech delay; non-verbal	Bilateral OA	West Syndrome; IS and FS; generalized tonic seizures	ASD traits	ND	5
DI16	LOVD: NR2F1_000057; CH16, #15; RE20, #11	De novo in-frame deletion in DBD	p.Phe110del	CC thinning; LV asymmetry; septum pellucidum agenesis; abnormal gyration	Yes	Severe; non-verbal	No or ND	West Syndrome; spasms at 3 mo; IS; hypsarrythmic EEG; electroclinical spasms	ASD	Global	6
CH16	LOVD: NR2F1_000041; CH16, #1; BE18, #16; RE20, #16	De novo MM in DBD	p.Cys128Arg	CC thinning; WM reduction; leukodystrophy	Yes, DMD	Yes (IQ ND); speech delay; non-verbal	OA; GVI	Epilepsy with staring spells and generalized tonic-clonic seizures	ASD; self-injurious behavior; head-banging	Yes	7
MI14; CH16	LOVD: NR2F1_000075; CH16, #2; BE18, #17; RE20, #17	De novo MM in DBD	p.Arg135Ser	CC thinning; WM reduction; ON malformation and OC bilateral hypoplasia	Yes, DMD	Yes (IQ ND); speech delay	Mild OA; ONH; CVI; GVI	IS; Occipital lobe epilepsy	ASD; head-banging	Profound, axial and appendicular	7
CH16; EL17	LOVD: NR2F1_000007; CH16, #3; BE18, #18; RE20, #19	De novo MM in DBD	p.Cys138Tyr	WM reduction	Global	Yes	OA; GVI	FS; Abnormal EEG during sleep	ASD; RB (persistent head-banging)	No	6
CH16	LOVD: NR2F1_000058; CH16, #4; BE18, #19; RE20, #21	De novo MM in DBD	p.Arg142Leu	CC thinning at 7 mo	Yes, DMD	Yes (IQ ND); speech delay; non-verbal	OA; P/SOD; small ON; CVI; GVI	IS; atonic seizures with markedly abnormal EEG	ND	Yes	6
CH16	LOVD: NR2F1_000010; CH16, #5; BE18 #20; RE20 #22	De novo MM in DBD	p.Cys146Arg	CC thinning and septo-optic dysplasia	Yes, DMD	Yes (IQ ND); speech delay; non-verbal	OA; GVI	No	ASD traits; RB (self-stimulatory behaviors)	Yes	6
CH16	LOVD: NR2F1_000024; CH16, #6; BE18, #21; RE20, #47;	De novo MM in DBD	p.Ala155Thr	Normal or ND	No	Mild speech delay (pronunciation; dysarthria)	No or ND	No	NO or ND	Yes	2
CH16	LOVD: NR2F1_000078; CH16, #7; BE18, #22; RE20, #51	De novo MM in LBD	p.Gly368Asp	Normal or ND	Yes	Yes (IQ ND); speech delay	No or ND	First generalized seizure at 18 yo	ASD; RB; aggressive behavior	No	4
CH16	LOVD: NR2F1_000057; CH16, #8; BE18, #23; RE20, #10	De novo in-frame deletion in DBD	p.Phe110del	CC thinning	Yes, DMD	Yes (IQ ND); speech delay; non-verbal	OA; ONH; CVI; GVI	IS	No or ND	Yes	6
CH16	LOVD: NR2F1_000053; CH16, #9; CH16, #24; RE20, #40	De novo frameshift truncation	p.Gly35Argfs*361	Normal or ND	Yes, DMD	Yes (IQ 55–69; verbal IQ 35–40 at 16 yo); speech delay	OA; P/SOD; CVI; GVI	Few seizures at 3–4 yo	ASD; RB including PDD-NOS at 22 yo; ADHD	Yes	6
CH16	LOVD: NR2F1_000055; CH16, #10; BE18, #25; RE20, #41	De novo frameshift truncation	p.His97Hisfs*22	ND	Yes	Yes (IQ ND); speech delay	OA; ONH; GVI	No	ASD; OCB/RB; ADHD	Yes	5
CH16	LOVD: NR2F1_000052; CH16, #11; BE18, #26; RE20, #36	De novo TIV	p?	Normal or ND	Yes, DMD	Yes (IQ ND); speech delay; non-verbal	OA; ONH; GVI	No	ASD; RB (head banging)	Yes	5
CH16	LOVD: NR2F1_000052; CH16, #12; BE18, #27; #37 in RE20, #37	De novo TIV	p?	CC thinning; ONH	Yes, DMD	Yes (IQ ND); speech delay; non-verbal	OA; ONH; P/SOD; CVI; GVI	Yes	ASD traits	Yes	7
CH16	LOVD: NR2F1_000048; CH16, #13; BE18, #28; RE20, #35	De novo TIV	p.M1?	ND	Yes, DMD	Yes (FSIQ in the 40 s); speech delay	OA; ONH; CVI; GVI	No	ASD; head banging	Substantial, central	5
CH16	LOVD: NR2F1_000048; CH16, #14; BE18, #29; RE20, #38	De novo TIV	p.M1?	CC and CS thinning; pyramidal decussation agenesis; right vs. left fiber directionality asymmetry	Yes, DMD	Yes (IQ ND); speech delay; non-verbal	OA; coloboma; ONH; GVI	Seizure at 3 yo, complex partial, left parietal	OCB (hand stereotypes); ADHD	Yes	7
CH16	LOVD: NR2F1_000042; CH16, #15; BE18, #30; RE20, #39	De novo TIV	p?	Cerebral malformations; bilateral HPM	Yes, DMD	Yes (IQ ND); speech delay; non-verbal	OA; GVI	Tonic-clonic seizures at 13 and 18 yo	OCB	Yes	7
CH16	CH16, #16; BE18, #31; RE20, #29	De novo deletion (0.2 Mb)	Deleted; (del. includes FAM172A, partial)	ND	Yes, DMD	Yes (IQ ND); speech delay	OA; P/SOD; pigmented maculae; GVI	No	ADHD	Yes	5
CH16	CH16, #17; BE18, #32; RE20, #31	Deletion (0.9 Mb)	Deleted; (del. includes FAM172A; KIAA0825, partial)	ND	Yes	Yes (IQ ND); speech delay	OA; P/SOD; GVI	No	ASD; ADHD	No	4
CH16	CH16, #18; BE18, #33; RE20, #32	Parental (son of CH16, #17), deletion (0.9 Mb)	Deleted	CC agenesis; DM of the EC and IC anterior limb; focal abnormality of the right CB	Yes, DMD	Yes (IQ ND); speech delay	OA; P/SOD; GVI	No	ASD traits; ADHD	Axial	6
CH16	CH16, #19; BE18, #34; RE20, #30	Deletion (1.2 Mb)	Deleted; (del. Includes FAM172A, KIAA0825, ANKRD31)	ND	Yes	Yes (verbal IQ 96; non-verbal IQ 70)	OA; GVI	No	ASD; PDD-NOS	No	4
CH16	CH16, #20; BE18, #35; RE20, #23	Deletion (5.0 Mb)	Deleted; (del. Includes FAM172A, KIAA0825, ANKRD32, MCTP30)	MCP	Yes	Yes (IQ ND)	No or ND	No	ND	Low muscle tone, normal mass and strength	4
KA17	LOVD: NR2F1_000039; BE18, #36; RE20, #1	De novo MM in DBD	p.Cys86Phe	CC thinning; WM reduction; MCP	Yes, DMD	Yes (DQ < 25 at 14 yo); speech delay; non-verbal	OA; mild bilateral ONH; CVI; GVI	One episode of IS; left occipital onset seizure in EEG; FS	Severe ASD; RB (self-stimulating, self-injurious behavior); limited social interaction	Yes	7
KA17	LOVD: NR2F1_000079; BE18, #37; RE20, #52	De novo MM in LBD	p.Leu372Pro	ND	Yes, DMD	Yes (IQ ND); speech delay	OA; GVI	ND	RB; ADHD	Yes	5
VI17	LOVD: NR2F1_000017	MM in DBD	p.Gly105Asp	Cerebral malformations	ND	Yes (IQ ND)	No or ND	ND	ND	ND	2
MH18	LOVD: NR2F1_000040; BE18, #38; RE20, #5	De novo MM in DBD	p.Lys96Glu	CC thinning	Yes	Yes (IQ ca. 30–50); speech delay	Mild OA; CVI; GVI	No	ND	Yes	5
PA19	LOVD: NR2F1_000038; RE20, #45; JU21, #10	Truncation	p.Tyr171*	CC thinning	Yes	Yes; mild (IQ ca. 77–80)	OA; GVI	No	Behavioral disorders; ADHD	ND	5
BO20	LOVD: NR2F1_000037; RE20, #43	De novo truncation	p.Gln28*	ONH	No	No (verbal IQ 141; nonverbal IQ 63)	OA; ONH; CVI; mild GVI	No; EEG showed rare isolated sharp waves from central regions	ASD (hand flapping and toe walking at 24 mo); behavioral disorders; ADHD	Yes	5
BE20	LOVD: NR2F1_000059; BE20, #1	De novo MM in DBD	p.Arg142His	CC thinning; OA (OC and nerve thinning); abnormal gyration	Yes	Yes	OA; amblyopia	IS at 8 mo	ASD and ADHD traits	Yes	7
BE20	LOVD: NR2F1_000060; BE20, #2	De novo truncation	p.Gln244*	CC thinning; ventricular asymmetry and enlargement; abnormal gyration; polymicrogyria	Yes	Yes	No or ND	ND	Behavioral disorders	Yes	5
BE20	LOVD: NR2F1_000054; BE20, #3; JU21, #4	De novo truncation	p.Glu39*	CC and OC thinning; CB malformation; ectopic nodular heterotopy; abnormal gyration	Yes	Yes (speech difficulties)	Severe bilateral OA; LVA	3–4 ES/y	Stereotypical movements; RB; ADHD	Yes	7
BE20	LOVD: NR2F1_000048; BE20, #4	De novo TIV	p?	CC thinning; cortical malformation; abnormal gyration	Yes	Yes (speech difficulties)	OA	ND	ASD and ADHD traits; behavioral disorders	No	5
BE20	LOVD: NR2F1_000056; BE20, #5	De novo MM in DBD	p.Tyr98His	CC thinning; OC hypoplasia; abnormal gyration	Yes	Yes	OA	ND	ASD traits; behavioral disorders; stereotypical movements	Yes	6
BE20	LOVD: NR2F1_000061; BE20, #6	De novo frameshift truncation	p.Lys323Serfs*73	Short CC; ON and chiasm thinning; hypoplastic olfactory lobes; abnormal gyration	Yes, DMD	Yes (speech difficulties)	OA; LVA	ND	ASD traits	No	5
ZO20	LOVD: NR2F1_000085	De novo truncation	p.Ser201*	ND	Mild/moderate	Mild/moderate	Bilateral P/SOD; LVA	ND	ND	ND	2
HO20	LOVD: NR2F1_000084; RE20, #44	De novo truncation	p.Glu85*	Normal or ND	Yes	Yes (IQ 69)	OA; GVI	Spells of behavioral arrest and non-responsiveness	ASD; auditory hallucinations and delusions; crying episodes	Yes	6
WA20	LOVD: NR2F1_000051	Frameshift truncation	p.Asn362fs*33	CC; ON and OC hypoplasia; mild MCP	Apparent at 8 mo	Speech delay	Severe GVI	Myoclonic astatic seizures at 2½ yo	ASD	ND	6
MI20	LOVD: NR2F1_000034; MI20, #1	De novo MM in DBD	p.Gly105Ser	Benign enlargement of the subarachnoid spaces (BESS)	Yes, DMD	Speech delay; non-verbal until 2 yo	Bilateral OA; GVI	Myoclonic epilepsy diagnosed at 3 yo	RB	ND	6
MI20	LOVD: NR2F1_000034; MI20, #2	De novo MM in DBD	p.Gly105Ser	LV enlargement; intraventricular arachnoid cyst	Yes, DMD	Speech delay; non-verbal until 2 yo	Bilateral OA; GVI	Myoclonic epilepsy diagnosed at 4 yo	RB	ND	6
ST20	LOVD: NR2F1_000035	MM in DBD	p.Lys107Glu	CC; ON; OC and optic tracts atrophy; complex pituitary cyst.	Marked and global; DMD	ND	Declining visual acuity; legally blind by 10 yo	1–3 yo + 30 episodes of FS; occasionally with myoclonus and generalized seizures	Aggressive behavior; depression; hallucinations	Yes	6
RE20	LOVD: NR2F1_000065; RE20, #33	De novo TIV	p.M1?	ND	No	ND	No or ND	No	ND	Yes	1
RE20	LOVD: NR2F1_000048; RE20, #34	De novo TIV	p.M1?	ND	Yes, DMD	Speech delay; non-verbal	OA; small ON; CVI; GVI	FS	ASD	Yes	5
RE20	LOVD: NR2F1_000067; RE20, #2	De novo MM in DBD	p.Cys86Arg	CC thinning	Yes	Speech delay; non-verbal	OA; CVI; GVI	IS	ASD traits	Yes	7
RE20	LOVD: NR2F1_000068; RE20, #3; RO20, #170	De novo MM in DBD	p.Val88Met	Normal or ND	Yes	Speech delay; non-verbal	OA; CVI; GVI	Onset at 9 wo; IS; focal and partial seizures; myoclonic jerks	ASD; RB (head banging)	Yes	6
RE20	LOVD: NR2F1_000009; RE20, #4	MM in DBD	p.Gly95Val	ND	Yes, DMD	Yes (IQ 56); speech delay	P/SOD; small ON; CVI; GVI	IS and absence seizures	ASD traits	Yes	6
RE20	LOVD: NR2F1_000069; RE20, #6	MM in DBD	p.Hys97Pro	Slightly decreased brain volume	Yes	Yes (IQ ND); speech delay; non-verbal	OA; CVI; GVI	Myoclonic seizures	ASD (severe)	Yes	7
RE20	LOVD: NR2F1_000070; RE20, #7	De novo MM in DBD	p.Tyr98Cys	Abnormal	Yes	Speech delay	P/SOD; ONH; CVI; GVI	Myoclonic; absence seizures	ASD; RB (head banging); ADHD	Yes	7
RE20	LOVD: NR2F1_000071; RE20, #8	De novo MM in DBD	p.Glu104Gly	ND	Yes	Speech delay; non-verbal	OA; CVI; GVI	No	ASD traits	Yes	5
RE20	LOVD: NR2F1_000072; RE20, #9	MM in DBD	p.Ser108Ile	ON thinning and small OC	Yes	Yes (IQ ND); speech delay; non-verbal	OA; small ON; CVI; GVI	No	ASD traits	Yes	6
RE20	LOVD: NR2F1_000073; RE20, #15	MM in DBD	p.Cys122Ser	ND	Yes, DMD	Speech delay; non-verbal	OA; GVI	IS	ASD traits	Yes	5
RE20	LOVD: NR2F1_000074; RE20, #42	frameshift truncation	p.Asn127Lysfs*270	ND	Yes, DMD	Speech delay	ONH; CVI; GVI	Yes	ASD; auditory hallucinations	Yes	6
RE20	LOVD: NR2F1_000076; RE20, #20	De novo MM in DBD	p.Gln139His	CC thinning; DM; ON thinning and small OC	Yes, DMD	Speech delay	OA; CVI; GVI	No	ASD	Yes	6
RE20	LOVD: NR2F1_000077; RE20, #49	MM in LBD	p.Ala311Pro	Normal or ND	Yes	Mild (FSIQ 80 below average); speech delay	P/SOD; mild GVI	Generalized Myoclonic and absence seizures	ASD	Yes	6
RE20		De novo MM in LBD	p.Glu318Asp	Abnormal	No but mild DMD	No (IQ 94; performance IQ 54)	OA; CVI; GVI	Atonic; Rolandic epilepsy	ASD	No	4
RE20	LOVD: NR2F1_000021; RE20, #46	De novo truncation	p.Arg373*	CC; ON and OC thinning	Yes, DMD	Mild (DQ ca. 60–70); speech delay	P/SOD; ONH; CVI; GVI	No	ASD	Yes	6
RE20	LOVD: NR2F1_000036; RE20, #54	De novo MM in LBD	p.Met406Thr	Small ON; Abnormal MRI	Yes, DMD	Yes (IQ ND); speech delay	CVI; GVI	No	ASD	Yes	6
RE20	LOVD: NR2F1_000063; RE20, #25	Maternal, deletion (2.5 Mb)	Deleted; (del. includes FLJ42709, FAM172A, MIR2277, POU5F2, KIAA0825, MIR1974, ANKRD32, MCTP1, FAM81B, TTC37)	ND	Yes	Yes (IQ ND); speech delay	OA; CVI; GVI	Absence and tonic seizures	ASD; PDD-NOS; OCD; pacing and hitting	Yes	5
RE20	LOVD: NR2F1_000064; RE20, #26	Deletion (0.97 Mb)	Deleted	ND	Yes	Speech delay	OA; small ON; CVI; GVI	No	ASD	Mild	5
JU21	JU21, #1	Frameshift truncation	p.Ala2Glnfs*3	CC thinning; abnormal gyration	Yes; mild	ND	OA; ONH; LVA	Occasional epileptic-like state during light sleep; FS at 4 yo; convulsions at 8 yo	ASD; ADHD	Hypotonia and hyperlaxity	6
JU21	JU21, #2	Frameshift truncation	p.Asn24Glyfs*379	Normal or ND	Yes	Yes (IQ ND); speech difficulties; learning disability	OA; ONH; CVI; LVA	No	ND	ND	3
JU21	JU21, #3	AA duplication	p.Arg31dup	Normal or ND	Yes	Yes (IQ ND); learning disability	OA; CVI; LVA	No	ND	ND	3
JU21	LOVD: NR2F1_000069; JU21, #5	De novo MM in DBD	p.Hys97Pro	CC; ON and OC thinning; periventricular leukomalacia; MCP	Yes; global (delayed visual maturation)	Yes (IQ ND); learning disability	OA; LVA	One episode of FS	ASD	Moderate/severe	7
JU21	JU21, #6	De novo truncation	p.Leu118*	CC mild foreshortening	Yes; global	Yes (IQ ND); learning disability	OA; LVA	No	ND	Yes	5
JU21	JU21, #7	(Likely) de novo truncation	p.Tyr120*	WM abnormalities	Yes; pervasive; global apraxia	Yes (IQ ND); speech delay; learning disability	OA; LVA	Myoclonic epilepsy; focal impaired awareness seizures	ASD; ADHD	ND	6
BA19; JU21	LOVD: NR2F1_000086; JU21, #8	De novo MM in DBD	p.Cys122Trp	WM reduction; CC thinning	Yes; global	Yes (IQ ND); learning disability	OA; CVI; LVA	IS; Myoclonic epilepsy	ASD; ADHD	Yes	7
JU21	JU21, #9	De novo MM in DBD	p.Ala155Thr	WM reduction; ON thinning	Yes	Yes (IQ ND); speech delay; learning disability	CVI; LVA	No	ASD; anxiety; limited attention span	ND	5
JU21	JU21, #11	De novo MM in LBD	p.Thr200Arg	Lateral and third ventricles enlargement; MCP	Global	Yes (IQ ND); learning disability	Central, steady, maintained	No	ND	Yes	5
JU21	JU21, #12	De novo truncation	p.Trp233*	CC; ON and OC thinning; WM delayed maturation; brain abnormalities	Yes	Yes (IQ ND); speech difficulties	OA; microphthalmia; small ON head; CVI	No	Limited concentration and short attention span	ND	5
JU21	LOVD: NR2F1_000082; JU21, #13	De novo MM in LBD	p.Glu342Lys	Normal CC e ON; OC atrophy and defective rotation; Normal gyration	No	No	OA; ONH; LVA	No	ND	ND	2
JU21	JU21, #14	(Likely) de novo deletion in LBD	p.Glu346_Gln349del	ND	Yes; walking delay	Yes (IQ ND); speech delay; dyslexia; learning disability	ONH; CVI; LVA	No	ND	ND	3
JU21	LOVD: NR2F1_000079; JU21, #15	Familial MM in LBD	p.Leu372Pro	ND	Yes; walking delay	Yes (IQ ND); speech delay	Small ON head; CVI; LVA	No	ND	ND	3
JU21	LOVD: NR2F1_000079; JU21, #16	Familial MM in LBD	p.Leu372Pro	Normal or ND	Yes; walking delay	Yes (IQ ND); speech delay;	OA; ONH; CVI; LVA	One episode of FS	ND	ND	4
JU21	LOVD: NR2F1_000079; JU21, #17	Familial MM in LBD	p.Leu372Pro	Normal or ND	Yes; walking delay	Yes (IQ ND); speech delay;	OA; CVI; LVA	No	ND	ND	3
JU21	JU21, #18	Familial deletion in LBD	p.Arg373_Leu374del	ND	No	ND	OA; ONH; LVA	No	ND	ND	1
JU21	JU21, #19	Familial deletion in LBD	p.Arg373_Leu374del	ON atrophy	No	ND	OA; ONH; LVA	No	ND	ND	2
JU21	JU21, #20	De novo MM in LBD	p.Gly395Ser	ON atrophy; WM loss	Yes	Yes (IQ ND); learning disability	CVI; LVA	No	ND	Generalized	5
JU21	LOVD: NR2F1_000083; JU21, #21	De novo truncation	p.Glu400*	CC thinning; ON and OC atrophy; abnormal gyration	Yes	Yes (IQ ND); learning disability	OA; ONH; LVA	No	ASD; behavioral disorders	ND	5
JU21	JU21, #22	De novo whole-gene deletion (599 Kb)	deleted; (del. includes FAM172A; NR2F1-AS1, partial; KIAA0825, last exon)	CC thickening; ON atrophy; cerebral vascular system abnormalities	Yes	Yes (IQ ND); mild speech delay; learning disability	OA; LVA	No	ND	ND	4
JS20	LOVD: NR2F1_000036	De novo MM in LBD	p.Met406Thr	DM	Yes	Severe (IQ ND); speech delay; non-verbal	OA; suspected ON dysplasia; GVI	Seizures from 4 mo	Short attention span	ND	6
GA21	LOVD: NR2F1_000048	De novo TIV	p.M1?	Brain abnormalities; ON; OC and optic tract hypoplasia	Yes	Severe (IQ ND)	Right iris and chorioretinal coloboma; small ON; bilateral P/SOD; LVA	EEG at 12 yo showed possible occipital seizures	ND	Yes	6

**Table 2 cells-11-01260-t002:** **List of BBSOAS main features**. Table resuming the main clinical features of BBSOAS patients with *NR2F1* haploinsufficiency; some of them are often present at birth (congenital), including hypotonia, nystagmus and oromotor dysfunction. The clinical features of BBSOAS are variable, and not every individual necessarily manifests all features. Further, the severity of the condition varies from one individual to the next. Abbreviations: ASD, autism spectrum disorder; ID, intellectual disability; IQ, intelligence quotient; MRI, magnetic resonance imaging.

BBSOAS Main Feature(s)	Clinical Description(s)
**Developmental delay (DD)**	Delay in reaching language, social or motor skills milestones
**Intellectual disability (ID)**	Significantly reduced ability to understand new or complex information and to learn and apply new skills (impaired intelligence). IQ ranging from profound ID with IQ < 20, to moderate (35 < IQ < 49) or mild ID (50 < IQ < 69)
**Visual impairment**	Optic nerve abnormalities and/or brain-based vision impairment:
	Optic nerve atrophy or pallor
	Optic nerve hypoplasia
	Cortical visual impairment (difficulty locating objects in a crowded field and following rapidly moving images and scenes).
	Alacrima (abnormal amount of reflex tearing)
	Manifest latent nystagmus and poor tracking; congenital
	Significant refractive errors
	Amblyopia
**Hypotonia**	Low muscle tone; congenital
**Oromotor dysfunction**	Swallowing, sucking and chewing problems; congenital
**Repetitive behavior**	Hand flapping, head banging and more
**Autism spectrum disorder (ASD)**	ASD or autistic traits
**Seizures**	Infantile and/or febrile; occipital seizures
**Attention-deficit hyperactivity disorder (ADHD)**	Inattention, impulsivity and hyperactivity
**Hearing impairment**	Abnormal hearing
**Dysmorphic facial features**	Mild and inconsistent
**Thin corpus callosum and neocortical dysgyria**	Hypoplasia of the corpus callosum and abnormal pattern of cortical convolutions and sulci (dysgyria in temporal and parietal areas) on brain MRI

**Table 3 cells-11-01260-t003:** **Clinical exams recommended for children with BBSOAS.** Table showing the main clinical tests for patients with *NR2F1* haploinsufficiency. Abbreviations: MRI, magnetic resonance imaging; OCT, optical coherence tomography; RNFL, retinal nerve fiber layer.

Recommended Clinical Exam(s)	Exam Description
**Developmental assessment**	Identify areas of impairment and allow for early intervention.
**Psychological evaluation for autism**	ADI-R (Autism Diagnostic Interview, Revised) and ADOS (Autism Diagnostic Observation Schedule) testing performed by a certified clinical psychologist.
**Brain MRI**	Recommended at age three years or older.
**Full, dilated eye examination**	Fundus photography and OCT scan of RNFL to document optic nerve health performed by an ophthalmologist every two years.
**Visual acuity tests**	As appropriate for patient’s age and understanding
**Full hearing evaluation**	Every two years

**Table 4 cells-11-01260-t004:** **Recommended therapeutic approaches.** Table resuming the recommended therapeutic approaches for BBSOAS patients. Abbreviations: ASD, autism spectrum disorder; CVI, cortical visual impairment.

Recommended Therapeutic Approach(es)	Therapy Description
**Visual therapy**	Focused on CVI
**Physical therapy**	Aiming to increase strength and to improve gross motor skills.
**Occupational therapy**	Aiming to improve fine motor skills and coordination.
**Speech therapy**	Consideration of sign language and alternative communication devices
**ABA (Applied behavioral analysis) therapy**	If ASD is diagnosed
**Anti-convulsive treatment**	If epilepsy is present

**Table 5 cells-11-01260-t005:** **Symptom prevalence: overall and by variant type.** Table shows the prevalence of specific morphological and pathological features (lines) in BBSOAS patients, as a whole (first column), or in distinct BBSOAS genetic categories (second to seventh columns), as indicated. The prevalence of each symptom is calculated as a percentage of positive cases above the total number of patients for that category (numbers of patients are indicated in parenthesis). The color code refers to the severity of prevalence, from higher incidence (red) to lower prevalence (blue). Abbreviations: DBD, DNA binding domain, LBD, ligand binding domain; TIVs, translation initiation variants; CC, corpus callosum; DD, developmental delay; DMD, delayed motor development; ID, intellectual disability; ASD, autism spectrum disorder; ADHD, attention deficit hyperactivity disorder; CVI, cortical visual impairment; OA, optic atrophy; ONH, optic nerve hypoplasia; P/SOD, pallid or small optic disc.

	Overall prevalence (N = 92)	Variants in the DBD (N = 32)	Variants in the LBD (N = 17)	Deletions (N = 15)	TIV (N = 9)	Truncations (N = 11)	Frameshift (N = 7)
**Average severity index**	4.94	5.62	3.76	4.33	5.33	5.18	5.29
**Phenotypic** **feature**							
**Morphology**							
Myelin defects	14.13% (13/92)	25.00% (8/32)	11.76% (2/17)	6.67% (1/15)	0.00% (0/9)	18.18% (2/11)	0.00% (0/7)
CC malformations	32.61% (30/92)	46.88% (15/32)	0.00% (0/17)	13.33% (2/15)	33.33% (3/9)	63.64% (7/11)	42.86% (3/7)
**Development and** **behavior**							
DD	88.04% (81/92)	90.62% (29/32)	70.59% (12/17)	93.33% (14/15)	88.89% (8/9)	90.91% (10/11)	100% (7/7)
DMD	30.43% (28/92)	40.63% (13/32)	11.76% (2/17)	20.00% (3/15)	66.67% 6/9)	9.09% (1/11)	42.86% (3/7)
ID/speech delay	86.95% (80/92)	93.75% (30/32)	70.59% (12/17)	86.67% (13/15)	88.89% (8/9)	90.91% (10/11)	85.71% (6/7)
ASD	38.04% (32/92)	40.63% (13/32)	29.41% (5/17)	26.67% (4/15)	33.33% (3/9)	45.45% (5/11)	71.43% (5/7)
ASD-like traits	14.13% (13/92)	28.13% (9/32)	0.00% (0/17)	6.67% (1/15)	22.22% (2/9)	0.00% (0/11)	14.29% (1/7)
ADHD	18.48% (17/92)	9.38% (3/32)	5.88% (1/17)	26.67% (4/15)	22.22% (2/9)	36.36% (4/11)	42.86% (3/7)
**Visual system**							
CVI	42.39% (39/92)	53.13% (17/32)	47.06% (8/17)	26.67% (4/15)	33.33% (3/9)	27.27% (3/11)	42.86% (3/7)
OA	67.39% (62/92)	78.13% (25/32)	47.06% (8/17)	53.33% (8/15)	77.78% (7/15)	72.73% (8/11)	71.43% (5/7)
ONH	21.74% (20/92)	12.50% (4/32)	29.41% (5/17)	0.00% (0/15)	44.44% (4/9)	27.27% (3/11)	57.14% (4/7)
P/SOD	19.56% (18/92)	18.75% (6/32)	11.76% (2/17)	33.33% (5/15)	22.22% (2/9)	18.18% (2/11)	14.29% (1/7)
**Others**							
Epilepsy	45.65% (42/92)	62.50% (20/32)	29.41% (5/17)	26.67% (4/15)	55.56% (5/9)	45.45% (5/11)	42.86% (3/7)
Hypotonia	61.96% (57/92)	75.00% (24/32)	35.29% (6/17)	60.00% (9/15)	88.89% (8/9)	54.55% (6/11)	57.14% (4/7)

## Data Availability

Not applicable.

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
