# Peer review of "Pathophysiological Heterogeneity of the BBSOA Neurodevelopmental Syndrome"

_cells, 2022, doi:10.3390/cells11081260_

Round 1

Reviewer 1 Report

1, Authors discussed BBSOAS genetic categories and NR2F1 function. Since it has been reported that point mutations lead to much more severe symptoms than deletion mutations. Especially, it seems that the molecular mechanisms of the former are much more complicated than the latter. Authors had better summarize more specific information about the severity index in intellectual disability (IQ) and visual impairment, two of the most common symptoms of BBSOAS between these two categories.

2, Cone photoreceptors respond to bright light for color perception and visual acuity. It has been demonstrated that Nr2f1 gene is required for the appropriate differentiation of cone photoreceptors (Satoh et al., 2009), which could be a possible cause for visual impairment observed in BBSOAS patients. The finding above should be added in the manuscript.

Satoh S, Tang K, Iida A, Inoue M, Kodama T, Tsai SY, Tsai MJ, Furuta Y, Watanabe S*. 2009 The spatial patterning of mouse cone opsin expression is regulated by bone morphogenetic protein signaling through downstream effector COUP-TF nuclear receptors. J. Neurosci. 29, 12401-12411.

Author Response

We would like to thank the reviewer for the insightful suggestions.

  1. We have now added additional information concerning the prevalence of main symptoms (intellectual disability, visual deficits such as optic atrophy and optic nerve hypoplasia, and epilepsy) in the appropriate sections of the text, thus enriching the genotype-phenotype correlation.
  2. We agree that while previous work focused mainly on retinal ganglion cells, Nr2f1 could also affect visual acuity by acting on the development of other cell types, such as photoreceptors (Satoh et al., 2009) or amacrine cells (Inoue et al., 2010). We have now added these two relevant papers in the section describing Nr2f1 role in retinal development.

Reviewer 2 Report

In this review, Bertacchi et al nicely summarized the clinical features of the recently described monogenic neurodevelopmental disorder the Bosch-Boonstra-Schaaf Optic Atrophy Syndrome (BBSOAS) and correlated the clinical features with in vivo and in vitro models with various NR2F1 mutations. As BBSOAS is extremely rare and has long been grouped with other diseases, the review serves as a comprehensive summary of the disease and demonstrated the correlation with NR2F1, which could be very useful for clinician scientists.

It is highly appreciated that the authors collected a tremendous number of clinical manifestations related to the disease and provided sufficient phenotypes of animal models. However, the scope of Cells focuses studies related to cell biology, molecular biology and biophysics. Although the authors have tried to correlate the phenotypes of patients and animal models with NR2F1, the information related to mechanisms, i.e., how NR2F1 mutations lead to such phenotypes, is still limited, and thus the manuscript might not be of interest for readers of Cells.

The authors also mentioned in the abstract “by modelling possible molecular mechanisms resulting from distinct NR2F1 genetic alterations, we try to set the path to future explorations of causative links between impaired brain development and subsequent symptoms in human neurological syndrome, by using the NR2F1 gene and BBSOAS as a paradigm of monogenic rare neurodevelopmental disorder”. As mentioned above, the “molecular mechanisms” of how NR2F1 mutations lead to distinct phenotypes were barely discussed in the review, as the review mainly focused on what phenotypes NR2F1 mutations lead to. In addition, BBSOAS seems extremely rare with more than 100 patients identified. With such rare number of patients, it is challenging to identify potential genetic modifiers to contribute to the pathophysiological heterogeneity. The authors did not mention about how to compensate for this shortcoming and how to use what we have known about NR2F1 mouse models to develop treatments for the disease.

Author Response

We thank the reviewer for appreciating our work and for the insightful comments.

  1. While we can agree that normally the scope of Cells is mainly related to cell biology, molecular biology and biophysics, we would like to clarify that the present review is part of a special issue dealing with the "Pathophysiological Mechanism of Neurodevelopmental Disorders". Hence, we believe the content of our review perfectly fits the scope of this specific journal issue. The largely unknown nature of molecular mechanisms regulating NR2F1 function and their consequences on clinical phenotypes demand for further experiments, and any speculative molecular mechanism or possible future therapeutic treatment we included in the text could be of interest for readers of Cells dealing with BBSOAS or, more in general, with neurodevelopmental diseases sharing common cellular and molecular pathways. We also would like to suggest the reviewer to read our recent review in "Front Mol Neurosci. 2021 Dec 15;14:767965. doi: 10.3389/fnmol.2021.767965" dealing with more molecular and cellular aspects of Nr2f1.
  2. As the Reviewer correctly pointed out, very little is known about the molecular pathways resulting from distinct NR2F1 genetic alterations. As our speculative revision of distinct NR2F1 mutations mainly deal with the efficiency in dimer formation, we decided to change the sentence by clearly stating that we are “modelling distinct NR2F1 genetic alterations in terms of dimer formation and nuclear receptor binding efficiencies, […] to estimate the total amount of functional NR2F1 acting in developing brain cells in normal and pathological conditions”.

    We also agree with the Reviewer that the scarcity of reported BBSOAS cases currently hampers the possibility to unravel the mechanisms underlying the clinical heterogeneity at a genetic and molecular level. We have now better stressed this point in the current version of the text (lines …: “The increase in number of patients would allow to explore the impact of distinct NR2F1 genetic perturbations at a cellular and molecular level, and how this contributes to their pathophysiological heterogeneity.”).

    Concerning the development of future therapeutic approaches, we further developed this aspect that – we agree with the Reviewer- was still weak. (See also response to Reviewer 3). However, and again, please keep in mind that, especially in the case of syndromic children with complex neurological conditions of genetic origin, therapeutic approaches are still symptomatic. Novel -more efficient- cures remain purely hypothetical to date.

Reviewer 3 Report

In my opinion, this is certainly relevant and needed review about a very rare disease. The review is well written, although is at times repetitive and is in general quite lengthy.

I suggest authors to make a few changes:

To add a list of all abbreviations or double-check that every abbreviation is defined first in the text.

I wish authors add a (short) section or paragraph describing what is known about NR2F1 (and also NR2F2). Including on what chromosome is this gene located, size of the gene, general information about protein, its size lenght etc. Also where is it located/expressed- on cellular and organ level. That would increase value of the review.

As this is a review that will be read by doctors, I also suggest authors expand on suggestions for clinical practitioners- when to suspect this syndrom in a patient, recommendations on diagnosis, and especially on suggestions on how to proceed when the diagnosis is done. 

The discussion part largely repeats what has been said previously in the manuscript, as the manuscript is rather long, I recommend shortening the discussion, making it more concise, and providing speculation on how the research and search for treatment could be done in the future.

I recommend publishing this review article after it has been rewritten according to my suggestions. I must apologize for not being able to give a more detailed review in these troubling times.

Author Response

We thank the referee for the appreciation of our review and the insightful suggestions on how to improve it.

    1. We have now added a list of all abbreviations at the end of the text and also double-check that every abbreviation is defined first in the text.
    2. Regarding the suggestion of giving more details on NR2F1 chromosomal location and general information about the protein, we have added this information in Figure 1, providing details about NR2F1 gene in an illustrated -more direct- way. Concerning NR2F1 expression, we again opted for a visual representation (Figure 1), taking advantage of NCBI protein database and integrating these data in Figure 1. However, we prefer not to add the same information about the homolog NR2F2, as this is out of the scope of the present review and is already described elsewhere in the same Journal (Polvani et al., Cells. 2019 Dec 31;9(1):101. doi: 10.3390/cells9010101.)

    3. Regarding recommendations on diagnosis and how to proceed when the diagnosis is done, we have now added a new paragraph describing the best approaches for BBSOAS diagnostic evaluation in clinical practice (see page 14).

    4. Finally, regarding the length, we have shortened some repetitive parts in the Discussion, as requested, and added instead new hypotheses for future treatments.

Reviewer 4 Report

This is an extensive, well-written narrative review on the Bosch-Boonstra-Schaaf optic atrophy syndrome (BBSOAS), a rare genetic disease mainly involving the central nervous system. Description of this rare condition was developed in its many aspects, from clinics to genetics and the possible mechanisms. The review is also well illustrated by useful figures and tables. My only comment is about the interpretation of the imbalance of excitation/inhibition (lines 466-467), since interneurons could promote seizures at least in the initial phase of the epileptic activity, as recently reported in Levesque et al. (2020, doi:10.3390/ijms21249391; 2021, 10.1016/j.neubiorev.2021.08.020, shown in Figs 2 and 3). This complex relationship makes the interpretation provided by authors not completely reliable.

Author Response

We thank the referee for the appreciation of our review and have now added the statement that besides an imbalance between excitation and inhibition, seizures could also be promoted by just a dysfunction of GABAergic interneurons, particularly in the initial phase of epilepsis. We have also added the suggested paper.

Round 2

Reviewer 1 Report

Good job for NR2F1 (COUP-TFI) and BBSOAS community.